# Hypothermic Protection in Neocortex Is Topographic and Laminar, Seizure Unmitigating, and Partially Rescues Neurons Depleted of RNA Splicing Protein Rbfox3/NeuN in Neonatal Hypoxic-Ischemic Male Piglets

**DOI:** 10.3390/cells12202454

**Published:** 2023-10-15

**Authors:** Christopher T. Primiani, Jennifer K. Lee, Caitlin E. O’Brien, May W. Chen, Jamie Perin, Ewa Kulikowicz, Polan Santos, Shawn Adams, Bailey Lester, Natalia Rivera-Diaz, Valerie Olberding, Mark V. Niedzwiecki, Eva K. Ritzl, Christa W. Habela, Xiuyun Liu, Zeng-Jin Yang, Raymond C. Koehler, Lee J. Martin

**Affiliations:** 1Department of Neurology, Johns Hopkins University School of Medicine, 558 Ross Building, 720 Rutland Avenue, Baltimore, MD 21205-2196, USA; 2Department of Anesthesiology and Critical Care Medicine, Johns Hopkins University School of Medicine, 558 Ross Building, 720 Rutland Avenue, Baltimore, MD 21205-2196, USA; jennifer.lee@jhmi.edu (J.K.L.); ekuliko1@jhmi.edu (E.K.); volberd1@jhu.edu (V.O.); niedzmark@yahoo.com (M.V.N.);; 3Department Pediatrics, Johns Hopkins University School of Medicine, 558 Ross Building, 720 Rutland Avenue, Baltimore, MD 21205-2196, USA; 4Department of Biostatistics and Epidemiology, Johns Hopkins University School of Medicine, 558 Ross Building, 720 Rutland Avenue, Baltimore, MD 21205-2196, USA; 5Department of Pathology, Johns Hopkins University School of Medicine, 558 Ross Building, 720 Rutland Avenue, Baltimore, MD 21205-2196, USA; 6Department of Neuroscience, Johns Hopkins University School of Medicine, 558 Ross Building, 720 Rutland Avenue, Baltimore, MD 21205-2196, USA; 7The Pathobiology Graduate Training Program, Johns Hopkins University School of Medicine, 558 Ross Building, 720 Rutland Avenue, Baltimore, MD 21205-2196, USA

**Keywords:** cell death continuum, corticostriatal projection, motor cortex, neocortical pyramidal neuron, neonatal encephalopathy, RNA-binding protein

## Abstract

The effects of hypothermia on neonatal encephalopathy may vary topographically and cytopathologically in the neocortex with manifestations potentially influenced by seizures that alter the severity, distribution, and type of neuropathology. We developed a neonatal piglet survival model of hypoxic-ischemic (HI) encephalopathy and hypothermia (HT) with continuous electroencephalography (cEEG) for seizures. Neonatal male piglets received HI-normothermia (NT), HI-HT, sham-NT, or sham-HT treatments. Randomized unmedicated sham and HI piglets underwent cEEG during recovery. Survival was 2–7 days. Normal and pathological neurons were counted in different neocortical areas, identified by cytoarchitecture and connectomics, using hematoxylin and eosin staining and immunohistochemistry for RNA-binding FOX-1 homolog 3 (Rbfox3/NeuN). Seizure burden was determined. HI-NT piglets had a reduced normal/total neuron ratio and increased ischemic-necrotic/total neuron ratio relative to sham-NT and sham-HT piglets with differing severities in the anterior and posterior motor, somatosensory, and frontal cortices. Neocortical neuropathology was attenuated by HT. HT protection was prominent in layer III of the inferior parietal cortex. Rbfox3 immunoreactivity distinguished cortical neurons as: Rbfox3-positive/normal, Rbfox3-positive/ischemic-necrotic, and Rbfox3-depleted. HI piglets had an increased Rbfox3-depleted/total neuron ratio in layers II and III compared to sham-NT piglets. Neuronal Rbfox3 depletion was partly rescued by HT. Seizure burdens in HI-NT and HI-HT piglets were similar. We conclude that the neonatal HI piglet neocortex has: (1) suprasylvian vulnerability to HI and seizures; (2) a limited neuronal cytopathological repertoire in functionally different regions that engages protective mechanisms with HT; (3) higher seizure burden, insensitive to HT, that is correlated with more panlaminar ischemic-necrotic neurons in the somatosensory cortex; and (4) pathological RNA splicing protein nuclear depletion that is sensitive to HT. This work demonstrates that HT protection of the neocortex in neonatal HI is topographic and laminar, seizure unmitigating, and restores neuronal depletion of RNA splicing factor.

## 1. Introduction

Hypoxic-ischemic (HI) encephalopathy (HIE) is caused by reduced brain blood supply and oxygen from birth asphyxia and occurs in about 1 to 3 infants in every 1000 live term births in America and Western Europe [1]. Nearly one million infants die worldwide each year from HIE after perinatal asphyxia [2]. Hypothermia (HT) is the only approved treatment for neonatal HIE in western countries [3]. However, approximately one-third of survivors who receive whole body or head cooling still have moderate-to-severe impairments in executive, visuospatial, and motor functioning, language, and emotional maturity years later [4,5,6]. Because these functions are forged neocortically [7,8,9], these childhood outcomes suggest that the effects of HT on the gyrencephalic cerebral cortex are less therapeutic than in subcortical brain regions or that different neocortical areas or cell layers are affected variably by the injury and HT. The neocortex indeed segregates into highly sensitive and less vulnerable regions after neonatal HI with peri-Rolandic and watershed patterns [10,11,12] and cortical laminar necrosis that is associated with increased risk for spasticity [13], although total neocortical damage can prevail with severe insult [14]. The idea of differential regional protection with HT is supported by studies of infants cooled for neonatal HIE with follow-up MRI [15]. MRI of cooled infants with HIE also suggests that the pattern of neocortical insensitivity to HT appears predictive of long-term epilepsy originating in watershed territories of the neocortex [16]. Experimentally, gyrencephalic animals are propitious for examining the organization and degeneration of the neocortex [17,18]. Neonatal piglets have neocortical selective vulnerability, including laminar necrosis, after asphyxic cardiac arrest [19,20]. Piglets (<2 weeks of age) with hypovolemic ischemia and normothermia (NT) or mild HT (1 h) treatments show reduced neuropathology in the deep layers of the temporal and occipital cortex with HT but not in the frontal or parietal cortex and not in the superficial cell layers of any cortical region [21].

HIE commonly causes neonatal seizures, even in cooled newborns [22,23]. Seizure burden in infants is associated with brain injury and adverse neurodevelopmental outcomes [24,25]. The seizures can be overt or subtle and clinically undetectable [26,27,28,29]. There is debate about seizures having additional direct damaging effects on the immature brain, separate from the primary underlying HIE that instigates seizures. Some clinical studies report that neonatal seizures are independently associated with poor outcomes by causing additional brain damage [25,30]. However, others have found that neonatal seizures do not exert deleterious impacts independent of the underlying HIE [31]. In monkey and pig models of neonatal HI, seizures seem to worsen neuropathology [19,20,32,33,34,35,36]. Discordant outcomes in clinical and experimental settings could have manifold attributes, including seizures not being uniform due to differences in species, genetic background, detection methods, gyrencephaly and network connectivity, metabolic depletion and recovery, and complexity and conservation of RNA splicing proteins [37,38]. Discerning whether seizures in gyrencephalic neonatal animals with HI injury independently add or contribute to brain damage or reflect the underlying severity of encephalopathy is important because the answers can instruct clinical management [39]. Commonly used anticonvulsants are not strongly proven in efficacy and may be harmful [40] with neurotoxic potential in neonatal animals, including nonhuman primates [41,42,43]. The molecular mechanisms for seizure vulnerability in HIE infants are also unclear. RNA splicing proteins have attracted attention in seizure biology. Rbfox family members splice some candidate genes associated with epilepsy and can be mutated in patients with epilepsy [38,44].

We developed a neonatal piglet model that combines global HI and HT with continuous EEG (cEEG) monitoring during survival for neuropathological assessment. We addressed several questions: (1) Does HT protect neocortical areas differentially after neonatal HI? (2) Are cEEG-confirmed seizures associated with worse neuropathology after neonatal HI? (3) Does HT after neonatal HI protect against seizures and neocortical neuropathology? (4) Is clear secondary seizure-related encephalopathy distinguishable from primary HI injury evidenced by topographical and laminar damage in the neocortex? (5) Is a seizure-related cytopathology phenotype in the neonatal neocortex divisible from the typically predominant ischemic necrosis? We also examined the localization of RNA-binding FOX-1 homolog 3 (Rbfox3/NeuN) because the gene for this RNA splicing protein is seizure-associated with loss-of-function mutations, causing neonatal and childhood epilepsy [38,44,45,46]. This work advances the understanding of the effects of HT on the cerebral cortex after HI and the relationships among neonatal seizures, neocortical neuronal injury, and neuromolecular pathology.

## 2. Materials and Methods

### 2.1. Neocortical Cytoarchitecture and Connectomics in Normal Piglets

The animal protocols were reviewed and re-approved on 6 June 2023 by the Institutional Animal Use and Care Committee of Johns Hopkins University (protocol number SW23M119). Neonatal Yorkshire piglets (2 to 5 days old, 1–2 kg, males) were used for descriptive normative neuroanatomy and tract tracing experiments to identify, using cytology and connectivity, the locations of the motor and somatosensory cortices. We wanted to verify these regions for placement of cEEG electrodes and neuropathological assessments. Naïve piglets (*n* = 3) were deeply anesthetized with pentobarbital (50 mg/kg) and phenytoin (6.4 mg/kg, SomnaSol) and, after thoracotomy and left myocardial puncture and aortic catheterization, ice-cold 100 mM phosphate-buffered saline (pH 7.4) was perfused (~2 L) for body exsanguination followed by freshly prepared 4% paraformaldehyde (PF) in 100 mM phosphate buffer (pH 7.4) for brain fixation (~4 L). Appropriate tissue fixation was judged by the stiffness of the body and immovability of the jaw and limbs. The piglets were decapitated, and the head was placed in 4% PF overnight. The following day, the calvarium was carefully removed by rongeur, and the brain was extracted from the skull base, placed in PF overnight, and then immersed in 20% glycerol for cryoprotection.

The brains were bisected mid-sagittally, and individual cerebral hemispheres were frozen and serially cut using a sliding microtome into 40 µm thick floating sections in either the sagittal, horizontal, or coronal plane. Every 10th section was mounted on a glass microscope slide for Nissl (cresyl violet, CV) staining to study the neocortical cytology. Every 11th section was used for cytochrome C oxidase enzyme histochemistry, as previously described [19,20,47,48], for visualization of neocortical mitochondrial metabolic activity in situ.

Other neonatal piglets (*n* = 7) were used for brain tract tracing experiments to confirm the locations of the somatosensory and motor cortices for cEEG electrode placement. Anesthesia was administered through a nose cone with 5% isoflurane and 50% nitrous oxide in 50% oxygen. The piglets were intubated, and the anesthetic was changed to 1.5–2% isoflurane and 70% nitrous oxide in 30% oxygen. Using a sterile technique, catheters were placed in the external jugular vein and femoral artery. An intravenous (iv) fentanyl bolus (20 µg/kg) was administered followed by an infusion rate of 20 µg/kg/h. Additional fentanyl boluses of 10–20 µg/kg were administered as needed to mitigate discomfort.

Each piglet was mounted securely in a stereotaxic frame (Kopf Instruments, Tujunga, CA, USA) in a flat skull position. The head surface was shaved and washed with chlorhexidine antiseptic scrub and 70% alcohol, draped, and then painted with povidone-iodine solution. A midline incision was made in the scalp, carefully avoiding damage to the vulnerable superior sagittal sinus below. The scalp was reflected with clamps. The fascia was cleared with a bone spatula to visualize surface landmarks on the skull for burr holes. Burr holes were made using a high-speed dental drill with a 0.5 mm bit. Stereotaxic coordinates were selected from the atlas of Salinas-Zeballos et al. [49]. Because this atlas uses a 3-day-old (1.15 kg) piglet as a reference, targeting adjustments were made based on the neonatal piglet size and brain neuroanatomical histology. The brain regional stereotaxic coordinates were: motor cortex (2 mm anterior to bregma, 1 mm lateral, 2 mm ventral to cortical surface), anterior somatosensory cortex (1 mm anterior to bregma, 10 mm lateral, 2 mm ventral to cortical surface), middle somatosensory cortex (at bregma, 17 mm lateral, 2 mm ventral to cortical surface), posterior somatosensory cortex (5 mm posterior to bregma, 15 mm lateral, 2 mm ventral to cortical surface), and striatum (10 mm anterior to bregma, 10 mm lateral, 15 mm ventral to cortical surface). The motor cortex was verified by evoking muscle contractions upon cortical stimulation, while the somatosensory cortex was verified by somatosensory-evoked potentials, as previously described [50]. We used a combination of tract tracers, including recombinant neurotropic viruses genetically encoding different fluorescent proteins, and non-genetic connectivity reporters such as wheat germ agglutinin-horseradish peroxidase (WGA-HRP) and FluoroGold (FG) [47,51,52]. The viruses used were adeno-associated virus-enhanced green fluorescent protein (AAV-eGFP, Vector BioLabs, Malvern, PA, USA), adenovirus-cyan fluorescent protein (AV-CFP, GenTarget Inc., San Diego, CA, USA), and lentivirus-red fluorescent protein (LV-RFP, Amsbio, Cambridge, MA, USA). All recombinant viruses drove the expression of their fluorescent protein under control of the cytomegalovirus promoter for high neural cell expression. Commercially prepared WGA-HRP (Vector Laboratories, Burlingame, CA, USA) and FG (Fluorochrome LLC, Denver, CO, USA) were used. A Hamilton microsyringe (blunt tip) was used for injections (2–5 µL). The tracers were used at the following concentrations: AV-CFP (1 × 10^9^ IFU/mL), AAV-eGFP (1 × 10^12^ genome copies/mL), LV-RFP (1 × 10^8^ IFU/mL), FG (5%), and WGA-HRP (20%). The craniotomies were sealed with bone wax, the scalp was sutured, and the piglets emerged from anesthesia and were extubated. Piglet survival times were 3 or 7 days, then they were perfusion-fixed with 4% PF, and their brains were prepared for frozen sectioning.

Tracer visualization in brain sections was achieved through direct fluorescence, immunohistochemistry, and enzyme histochemistry [47,51,52]. eGFP, RFP, and FG were observed by their fluorescence. Highly specific antibodies were also used for the detection of RFP (BioVision, Walthan, MA, USA), FG (Millipore, Burlington, MA, USA), and CFP (BioVision). WGA-HRP was detected by enzyme histochemistry [53] with neutral red counterstaining.

### 2.2. Neonatal Piglet Model of Encephalopathy

Neonatal Yorkshire piglets (2 to 3 days old, 1–2 kg, males) were randomized to one of four groups: sham normothermia (NT), sham HT, HI-NT, or HI-HT. Unanesthetized unoperated piglets served as the naïve control group. The group sizes were sham-NT (*n* = 6), sham-HT (*n* = 10), HI-NT (*n* = 8), HI-HT (*n* = 10), and naïve (*n* = 6).

Anesthesia was induced with 5% isoflurane and 50% nitrous oxide in 50% oxygen delivered by a nose cone. After intubation, the anesthetic was changed to 1.5–2% isoflurane and 70% nitrous oxide in 30% oxygen. Sterile catheters were placed in the external jugular vein and femoral artery. A fentanyl bolus (20 µg/kg iv) was administered followed by infusion at a rate of 20 µg/kg/h. Additional fentanyl boluses (10–20 µg/kg) were administered as needed for discomfort.

Some piglets received craniotomies for placement of sterile, epidural 4-lead electrode telemetry arrays for cEEG (Stellar Telemetry, TSE Systems, Inc., Chesterfield, MO, USA). The placement (LJM, NRD) of the Stellar Telemetry real-time continuous recording electrode arrays (Figure 1A–C) was determined from previous neuroanatomical mapping of the topographic distribution of neocortical pathology in newborn piglets that survived for 1–4 days after HI [19,36] and key confirmatory neuroanatomical experiments shown here. The EEG recording arrays had 4 bipolar channels with each electrode attached securely to a miniature cranial screw inserted epidurally through a burr hole made at stereotaxically determined coordinates. Each electrode–epidural screw assembly was secured firmly to the surrounding bone by low-heat, quickset acrylic cement (Figure 1A–C). The immobility of each electrode was confirmed with forceps. The placement was bilaterally symmetrical in the right and left hemispheres. The anterior-most channel (designated right or left hemisphere anterior) had electrode placement 15 mm anterior to bregma and 3 mm lateral to the midline and was paired with an electrode placed 10 mm anterior to bregma and 6 mm lateral to the midline. For reference, bregma is identified in Figure 1A,D,E. The anterior channel recorded EEG signals from the anterior primary motor cortex and anterior primary somatosensory cortex. The posterior-most channel (designated right or left hemisphere posterior) had electrode placement 8 mm anterior to bregma and 3 mm lateral to the midline and was paired with an electrode 4 mm anterior to bregma and 6 mm lateral to the midline. The posterior channel recorded EEG signals from the posterior motor cortex and posterior primary somatosensory cortex. The electrode array transmitter (Figure 1C) was inserted subdermally in a nape pocket. The left- and right-side electrode wire leads were loosely braided and flanked laterally under each side of the scalp. The array antenna was secured to the posterior-most point of the scalp incision with sutures and emerged externally. The surgical incision was closed with 3.0 suture. Upon completion of surgery, the isoflurane was discontinued, and the fentanyl was decreased to 10 µg/kg/h. All piglets received vecuronium at 0.2 mg/kg/h to prevent ventilatory efforts during hypoxia-asphyxia and shivering during HT. The same anesthetic regimen was administered to all HI and sham piglets. The cortical electrode placement for each piglet was confirmed visually (LJM) after brain perfusion fixation (Figure 1D,E).

### 2.3. Global HI Injury and Mild HT

The inhaled oxygen was decreased to 10% for 45 min. Then, room air was supplied for 5 min to reoxygenate the heart and improve cardiac resuscitation in our model. The endotracheal tube was then clamped to induce asphyxia for 8 min [54,55,56]. With this protocol, the piglets developed severe bradycardia and hypotension with heart rates <60 beats per minute or mean arterial blood pressure (MAP) less than 45 mmHg. The piglets were resuscitated with an inhaled oxygen concentration of 50%, manual chest compressions, and epinephrine (100 µg/kg iv). Piglets that were not resuscitated within 3 min of chest compressions were excluded. Sham piglets received 50% inhaled oxygen for 3 min. After return of spontaneous circulation (ROSC) or the time equivalent in sham piglets, 70% nitrous oxide in 30% oxygen was restarted.

NT in neonatal piglets was a rectal temperature in the range of 37.5–39.0 °C and was maintained with a heating blanket [57,58,59]. In piglets randomized to receive HT, whole body HT was started 2 h after ROSC using ice packs and a cooling blanket to a goal rectal temperature of 34 °C [54]. This change in temperature is like the 4 °C decrease in clinical HT [60]. HT was initiated 2 h after resuscitation to mimic clinical delays in cooling [61].

Three hours after ROSC, ketamine (10 mg/kg/h iv) was started, and nitrous oxide was decreased to 33% in 33% oxygen and 33% air. For the remainder of the anesthetic, fentanyl and ketamine were increased in 10 µg/kg/h and 10 mg/kg/h increments, respectively, with additional boluses of fentanyl (10 µg/kg) and ketamine (10 mg/kg) administered as needed for comfort. Dopamine was given when needed to maintain the MAP above 40 mmHg during the overnight HT or NT protocols.

Rewarming to NT began in HT piglets at 20 h from onset by increasing the temperature of the water circulating through the blanket. The rate of rewarming was 0.5 °C/h, which is the clinical rewarming rate at the Johns Hopkins Hospital NICU [60]. Piglets reached their NT target temperature of 38.5 °C at ~29 h from onset. Vecuronium infusion was stopped 15 h after ROSC to allow time for the neuromuscular blockade to wear off before extubation. Piglets then emerged from anesthesia and were extubated. After regaining postural and ambulatory control, the piglets were returned to their cages with free access to milk and continued overnight supervision and intermittent video monitoring (LJM, Appendix A).

### 2.4. Piglet Survival and Perfusion Fixation

The piglets survived for 2–7 days. They did not receive anticonvulsant medications. Humane treatment and the survivability of the piglets required their independent ambulation and feeding (drinking milk) ad libitum. Piglets that became non-ambulatory and unable to drink, had sustained general tonic-clonic seizures >20 min (Appendix A), or were deemed to be in status epilepticus [62] were euthanized before the 7-day endpoint. All piglets in this cohort had their brains prepared optimally for neuropathology. There were no postmortem delays before animal perfusion fixation. Piglets were euthanized with SomnaSol iv. Sham piglets were randomly selected to be euthanized at the same time (with the same solutions) as HI piglets throughout the 2-to-7-day period. After deep anesthesia, but prior to fatal cardiorespiratory arrest, the cEEG electrode arrays were quickly removed from the skull before perfusion fixation with 4% PF (LJM). All piglets were uniformly fixed. Afterwards, they were decapitated, and the head was immersed in 4% PF overnight. The following day, the calvarium was removed (LJM) and the brain with intact dura mater was examined and photographed (Figure 1D). Afterwards, the dura was carefully removed, and the brain was scrutinized for cortical damage and photographed (Figure 1E). This protocol was strictly followed to determine whether the cEEG electrode cranial screws were only epidural without underlying neocortical parenchymal damage (Figure 1D,E). The brain was then removed from the skull base and returned to PF overnight. The following day, the brain was immersed in 20% glycerol. Each brain, left and right cerebral hemispheres, was blocked in the coronal plane (Figure 2A–C) from the frontal lobes to the hindbrain, including the telencephalon, diencephalon, midbrain, pons, and medulla with cerebellum, and paraffin-processed in tissue cassettes. The paraffinized brain blocks were cut using a rotary microtome into 10 µm sections and mounted on glass slides for hematoxylin and eosin (H&E) staining (Figure 2) and immunohistochemistry (Figure 3).

### 2.5. H&E Neuropathology and Cell Counting

H&E-stained brain sections, the standard for clinical HIE postmortem neuropathological diagnosis [63], were used for neuropathological assessments. The assessments were performed blinded to piglet treatment. An investigator (JKL) used one H&E-stained section from each block for neuronal counting, including: (1) the frontal lobe with the dorsomedial anterior motor cortex and gyrus rectus and the dorsolateral and inferior prefrontal cortex (Figure 2A,D); (2) the anterior striatum and overlying central motor cortex and anterior primary somatosensory cortex (Figure 2B,E); and (3) the mid-thalamus with the overlying posterior motor and somatosensory cortices (Figure 2C,F). In separate piglets, the identities of these neocortical regions were confirmed by cytology and chemo- architecture (Figure 4; Appendix A) and connectivity (Figure 5). The primary somatosensory and motor regions were specifically assessed for neuropathology because they showed reproducible vulnerability to HI in many variations of this swine model of asphyxic cardiac arrest [19,20,36,56,64], and these brain regions in piglets had high endogenous mitochondrial metabolism (Figure 4; Appendix A), sodium/potassium ATPase activity [19,20], and blood flow [65]. Another investigator (LJM) counted the inferior parietal neocortex for neuropathology (Figure 2B,E). This area was assessed because previous topographic mapping suggested that this region in piglets is typically not vulnerable to the primary HI insult, but in the presence of clinical seizures, damage spreads into this area of the neocortex [19,20].

A profile counting-based approach was used for quantification of the amount of neuronal damage in each piglet brain. Individual neuron profiles were counted microscopically in the medial and lateral banks of the anterior and mid-parietal somatosensory cortex and middle and posterior motor cortex. The anterior motor cortex was counted medial to the dorsolateral prefrontal cortex (Figure 2D). The identification of these regions was achieved by consensus between two investigators (JKL, LJM). At 400× magnification, five vertical rows of microscope fields spanning cortical layers II through VI in each gyral bank were counted independent of laminar specification (Figure 2D–F). In the inferior parietal cortex, the number of morphologically normal neuron profiles was counted (LJM) in 10 horizontally arranged 1000× microscopic specifically in layers II, III, and V of the superior bank. These counts were performed at high magnification for careful assessment of neuronal cell death phenotype [66]. Additionally, 10 microscope fields at 400× were counted (JKL) in the putamen gray matter (Figure 2B) to assess the efficacy of HT protection, as previously described [54,64,67,68].

Counted neuronal profiles in H&E-stained sections were classified by their microscopic appearance [54,56,66,69]. Normal neurons (Figure 2G) were 8–15 μm in diameter, had a non-vacuolated cytoplasm, interpreted as intact membranous organelles without swelling, and an open nucleus (not condensed, darkly basophilic, or pyknotic) with at least one nucleolus and gossamer chromatin strands dispersed in a finely particulate nucleoplasmic matrix. Other neurons (Figure 2H) were unlike normal neurons, with a stark basophilic cytoplasm, few vacuoles or dilated cisterns, notably at a perinuclear location, seemingly intact cytoplasmic and nuclear membranes, and a prominent darky basophilic (blue-dark purple) nucleoplasmic matrix but a non-pyknotic nucleus with a nucleolus [54,56]. Because neurons typified by enhanced basophilia might survive [70,71] and correspond to neurons with enhanced chromatin-DNA template activity and RNA synthesis [72], they were combined into the total normal neuron category for analysis. Ischemic-necrotic neurons had a hematoxylin (blue-purple)-stained, angular, and pyknotic nucleus, angular soma, vacuolated and eosinophilic (red-pink) cytoplasm, and absence of perinuclear pallor (Figure 2I). These cells underwent dissolution of the plasma and nuclear membranes (Figure 2I) and showed nucleoplasmic matrix speckling [54,69]. Cells undergoing the apoptosis–necrosis continuum had ≥4 nuclear fragments of irregularly shaped chromatin clumps, basophilic or eosinophilic cytoplasm, some cytoplasmic vacuolation, but seemingly intact cell membranes (Figure 2J) [54,69]. Apoptotic cells could be identified as neurons because of their size and residual cytoplasm (Figure 2K) or they were cell type non-identifiable, had round and small profiles with eosinophilic, condensed cytoplasm, chromatin clumps (≤4 crescent-shaped or round clumps), and the cell surface was often withdrawn from the surrounding neuropil [54,69]. These cell degeneration morphologies were previously shown to be resolvable by H&E staining with their nuclear and cytoplasmic appearances weighted prominently in their divisibility [54,56,64,66,69].

### 2.6. Rbfox3 Immunohistochemistry (IHC)

Immunoperoxidase IHC with diaminobenzidine (DAB) as chromogen was performed on piglet brain paraffin sections as previously described [48,73,74,75,76] to localize Rbfox3 (previously known by its antibody, called NeuN) using a mouse monoclonal antibody (Millipore, Clone A60, Burlington, MA, USA ). This antibody was characterized for specificity in pig brain homogenates using Western blotting (Figure 3A, inset; Appendix A), and the results were similar as those described for human and mouse brains [77,78]. Nissl counterstaining with CV was performed for cellular and laminar identification in the neocortex and profile counting of total neurons.

The Rbfox3-stained brain sections were used for cell counting in the somatosensory cortex (Figure 3; Appendix A). Counting was performed specifically in layers II, III, and V. CV counterstaining defined layer identification by cell morphology, distribution, and packing density, as well as other landmarks, such as distances from deep layer I (a neuron-poor layer in piglets) and the very distinctive polymorphic layer VI just above the subcortical white matter (Figure 3A–C). After layer identification based on positive cell morphology and packing density (Figure 3A), neuronal counts were performed in 1000× non-overlapping microscopic fields distributed horizontally and intralaminarly (Figure 2E). Total neurons were counted based on CV counterstaining in each microscopic field, and then neurons were classified as normal Rbfox3-positive with nuclear and cytoplasmic staining (Figure 3B inset), damaged ischemic-necrotic or continuum Rbfox3-positive with pyknotic and clumped nuclear immunoreactivity (Figure 3C inset), and Rbfox3-depleted with depleted nuclear Rbfox3 immunoreactivity but otherwise appearing morphologically normal based on CV staining (Figure 3E).

### 2.7. EEG and Video Analysis

cEEG data were acquired for up to 7 days after HI (Stellar Telemetry, TSE Systems, Inc.). The EEG data were analyzed using Notocord-hem software (NOTOCORD Systems, Version 4.4.0.3, 2020, Philadelphia, PA, USA). An investigator (CTP, senior neurology resident) with experience in reading human EEGs analyzed the piglet EEGs with the guidance of two American Board of Psychiatry and Neurology epileptologists (EKR, CWH) who were experienced in analyzing human neonatal and experimental EEGs. These investigators were blinded to the treatment group and clinical video recordings. Electrographic seizure activity was defined as waveform activity with features of epileptiform pattern (spike wave, polyspikes, sharp wave, rhythmicity) of at least 10 s with confidence in evolution of waveform in frequency, morphology, or location. Seizure activity that clustered in time was counted as a single seizure if clustering occurred within 60 s of an event. Seizure start times, stop times, and waveform characteristics, including morphology, frequency, and focality, were documented within each recording. The seizure burden was defined as the percentage of the total EEG recording with seizure activity. Another investigator (LJM) who video recorded piglet behavior during recovery used the recordings to corroborate electrographic seizures and rule out motion artifacts. Seizures were allowed to self-resolve without anti-seizure medications in order to identify the natural biology of brain injury with seizures, but piglets with continuous clinical seizure activity (Appendix A) were euthanized.

### 2.8. Sample Size and Statistical Analysis of Data

A prior report [64] of neonatal piglets with HI injury that received 29 h of anesthesia, as in the current study, showed that the mean difference in ischemic-necrotic neurons within the neocortex between HI-NT and sham-NT piglets was 100 with a within-group standard deviation of 5. A sample size of 4 generated a power >0.9. We increased the sample size to allow for some variability in our estimates. All data were analyzed using GraphPad Prism 9 or XLSTAT 2023.1.5 software. A *p*-value < 0.05 was deemed significant.

Arterial blood pH, partial pressure of carbon dioxide (PaCO_2_), partial pressure of oxygen (PaO_2_), MAP, hemoglobin, and glucose levels at baseline and 1, 3, 20, 24, and 28 h after ROSC were analyzed by 2-way repeated measures analysis of variance with post-hoc Tukey tests. The blood gas and MAP data for 42 min of hypoxia and 7–8 min of asphyxia between HI piglets destined to receive NT or HT were compared by *t*-tests. Blood gas data were analyzed from 7 min of asphyxia and MAP data were analyzed from the 8 min time point of asphyxia. The physiology data are graphed as means with 95% confidence intervals. Survival duration was analyzed by Kruskal–Wallis analysis of variance by ranks test.

The differences between mean ratios of normal-to-total neurons and ischemic-necrotic-to-total neurons and numbers of apoptotic and apoptosis–necrosis continuum cells within the frontal, motor, and somatosensory cortices determined from H&E-stained sections were initially estimated using linear mixed models with random intercepts to account for measurements within the same piglet (33 comparisons for each cell category). Sham-NT was the reference group, and these analyses were adjusted for survival duration.

Neuropathology data were also analyzed separately outside of the linear mixed model. Ratios of ischemic-necrotic-to-total neurons and normal-to-total neurons, numbers of apoptotic and apoptosis–necrosis continuum cells, and numbers of normal neurons (cells/mm^2^), total neurons, and NeuN (Rbfox3)-depleted neurons in the somatosensory cortex were analyzed by one-way analysis of variance followed by independent sample *t*-tests with Bonferroni correction because of the design of planned orthogonal comparisons among groups and the calculated standard deviation of the different groups could be very dissimilar. However, *p*-value discovery was very similar with this approach compared to multiple comparison analysis.

The difference in seizure burden between sham and HI piglets was analyzed by Mann–Whitney tests. Spearman correlations were used to evaluate the relationship between seizure burden and the ischemic-necrotic-to-total neuron ratio. These data are graphed as box or scatter plots.

## 3. Results

### 3.1. Neonatal Piglet Gyrencephaly, Cytoarchitecture, Chemoarchitecture, and Connectivity

Like the human brain [79,80,81], the piglet brain has distinct cortical gyral and sulcul patterns that segregate into functional localizations, including the motor, somatosensory, and visual cortices (Figure 1E, Figure 2A–F and Figure 4A,E,F). These cortical functional localizations in piglets have been identified by electrophysiology [50,82,83] and fMRI-BOLD (P Liu, D Jiang, D Liu, LJ Martin, JK Lee, unpublished observations). As shown histologically by Nissl and Rbfox3 immunostaining, the motor cortex was discernable by prominent layers V and VI and less prominent layer IV typical of the agranular neocortex (Figure 3B and Figure 4B). In contrast, the somatosensory cortex was distinguished by its prominent layer IV and less prominent layer V (Figure 3A and Figure 4C,D). In piglets, as in humans [84], the cytological transition between the motor and somatosensory cortices can be identified (Figure 4D).

Nissl- and H&E-based histological frames of reference for the parcellation of the neonatal piglet neocortex were confirmed by in situ (tissue section) metabolic mapping for complex IV cytochrome C oxidase enzyme activity (Figure 4E–G, Appendix A) [20]. This assay showed that mitochondrial activity varied throughout different regions of the piglet neocortex. The highest activity levels in the neocortex appeared in the visual (Figure 4E), somatosensory (Figure 4F), and inferior frontal (Figure 4G) cortices. In these and other cortical regions, high cytochrome C oxidase activity was seen directly in neuronal cell bodies and the neuropil (likely synaptic mitochondria) in specific cortical layers in specific patterns (Appendix A). This work helped to consolidate our knowledge of the piglet neocortex and its compartmentation.

The connectivity experiments supported our identification of the motor and somatosensory cortex in neonatal piglets (Figure 5). Viral tract tracers were injected unilaterally into different neocortical regions (Figure 5A), while FG was injected unilaterally and subcortically into the striatum (Figure 5B). FG retrogradely labeled corticostriatal projection neurons in layer V of the motor cortex were observed (Figure 5B inset). Injections of AV-CFP robustly transfected pyramidal neurons in layer V of the motor cortex (lateral bank), as visualized with antibody to CFP (Figure 5C), including human Betz-like neurons (Figure 5C inset), thus confirming the primary motor cortex identity of the cortical area that was studied here neuropathologically. Pyramidal neuron dendrites originating from neurons in layer V could be visualized spanning the entire width of the cortex to layer I (Figure 5D). In the central putamen, single axons from motor corticostriatal projection neurons could be identified (Figure 5E). LV-RFP injections into the somatosensory cortex (Figure 5A,G,H) transduced neurons in the lateral bank and revealed that somatosensory corticostriatal projection axons traveled in the external capsule to target the lateral putamen (Figure 5F). Antibody detection of RFP revealed the fine terminal fields of corticothalamic projections to the ventral posterolateral thalamic nucleus (Figure 5H,I). AAV-eGFP injections into the somatosensory cortex showed, by retrograde labeling, extensive cortico-cortical connections with the primary somatosensory cortex, including layer V of the posterior cingulate cortex (Figure 5J) and layer III of the primary olfactory cortex. Injections of WGA-HRP into the posterior somatosensory cortex confirmed the lateral external capsule trajectory of corticostriatal preterminal axons, their entry into the lateral putamen, and extensive deposition of synaptic terminals in the central putamen (Figure 5L). Retrogradely filled neurons in the ventral posterolateral nucleus of the thalamus were also seen with WGA-HRP (Figure 5M), thus confirming the primary somatosensory identity of the cortical area that was studied here neuropathologically.

### 3.2. Neonatal Piglet Brain Injury Model

Forty piglets were used for the HI experiments. There was an 85% protocol completion rate with survival (see Appendix A: Animal Attrition and Appendix A). A total of 6 sham-NT, 10 sham-HT, 8 HI-NT, and 10 HI-HT piglets completed the protocols and were analyzed for histologic brain injury.

#### 3.2.1. Pathophysiology

Both HI groups had decreased PaO_2_ during hypoxia and asphyxia (Figure 6A, Appendix A) and corresponding increased PaCO_2_ during asphyxia (Figure 6B, Appendix A). Both HI groups had severe hypotension (<40 mm Hg) and arterial blood acidosis (pH < 6.9) during asphyxia (Figure 6C,D).

Target hypothermic core temperature was achieved in 3 h (Figure 6E). Rewarming from this target temperature began at 20 h of cooling and continued through 29 h for emergence from anesthesia and extubation. MAP and arterial blood pH did not differ significantly among the treatment groups at any time point during HT. Time and treatment interactively affected PaCO_2_ (*p *= 0.006). Post-hoc tests showed that sham-HT piglets had higher PaCO_2_ than HI-NT piglets at 1 h from cooling onset (*p* = 0.038; Figure 6B). PaO_2_ also varied across time (*p <* 0.001) with higher 1-h PaO_2_ in HI piglets than in sham piglets that would later receive HT (*p* = 0.049; Figure 6A). Hemoglobin levels (Appendix A) varied with time (*p* < 0.001). HI-HT piglets had higher hemoglobin levels than HI-NT piglets at 3 h (*p* = 0.029) and 20 h (*p* = 0.002) of HT (Appendix A). Hemoglobin levels did not differ between groups during NT or rewarming. Glucose level (Appendix A) was interactively affected by time and treatment (*p <* 0.001). At 1 h from onset of cooling (Appendix A), the HI group had higher glucose levels than the sham group that would receive HT (HI-NT: *p* = 0.013; HI-HT: *p* = 0.042). Sham piglets at 20 h of HT had higher glucose than sham-NT piglets (Appendix A, *p* = 0.048) and HI-NT piglets (*p* = 0.034). Glucose levels were also higher in HI-HT piglets than in HI-NT piglets at 20 h (*p* = 0.047). During rewarming at 24 h, the HI-HT piglets had higher glucose levels than HI-NT piglets (*p* = 0.018). Glucose levels did not differ among groups at 28 h and prior to emergence from anesthesia.

#### 3.2.2. HT Strongly Protects the Primary Somatosensory Cortex after HI

H&E-stained paraffin sections (10 µm thick) of the brain were used for neuron profile counting (Figure 2). The survival times of piglets for neuropathological assessment did not differ among treatment groups (*p *= 0.078; Appendix A).

HI-NT piglets had lower (*p* < 0.001) normal-to-total neuron ratios in the medial and lateral gyral banks of the somatosensory cortex within the anterior parietal cortex (Appendix A) relative to sham-NT piglets. The ratio of normal-to-total neurons in the anterior part of the somatosensory cortex in HI-HT piglets was significantly greater than that in HI-NT piglets in the medial (*p* = 0.002) and lateral (*p* = 0.01) banks but was not different from that in sham-NT piglets (Appendix A). Loss of normal neurons also occurred in the mid-parietal somatosensory cortex (Appendix A) in both gyral banks (medial: *p *= 0.001; lateral: *p* = 0.001). The ratio of normal-to-total neurons in the mid-parietal somatosensory cortex of HI-HT piglets was significantly greater than that of HI-NT piglets in the medial (*p* = 0.012) and lateral (*p* = 0.02) banks but was not statistically different from that of sham-NT piglets (Appendix A).

The phenotypes of neocortical neurodegeneration in HI piglets were examined by H&E staining. Classic ischemic necrosis (Figure 2I) was the predominant cell death phenotype seen in the neocortex compared to apoptosis and hybrid continuum death (Figure 2J,K and Figure 7, Appendix A). Neuronal ischemic necrosis was present in layers II–VI if the neocortical injury pattern was panlaminar (Figure 7F), or in layer II and upper layer IV (with patchy injury in layer 2) if the injury pattern was laminar (Figure 7E), as previously described [19]. This dominant presentation of neuronal ischemic necrosis (Figure 2I) was also seen in the putamen [48,56,73]. HI-NT piglets had very high proportions of ischemic-necrotic neurons in the somatosensory cortex relative to sham-NT piglets (Figure 7A–D, Appendix A). In the anterior parietal somatosensory cortex of HI-NT piglets (Figure 7A,B), the ratios of ischemic-necrotic-to-total neurons in the medial (*p *= 0.001) and lateral (*p *= 0.012) gyral banks were also higher than in HI-HT piglets. HI-NT piglets also had more ischemic necrosis in the medial (*p *= 0.003) and lateral (*p *= 0.025) banks of the somatosensory gyrus in the mid-parietal cortex than HI-HT piglets (Figure 7C,D). Interestingly, HT appeared to have differential protective potential in different neocortical regions (Figure 7), as the posterior somatosensory cortex medial bank appeared to be most significantly protected by HT. The amount of neuronal ischemic necrosis in different somatosensory cortical regions in HI-HT piglets did not differ significantly from that in both sham groups (Figure 7A–D).

Apoptosis (Figure 2K) and neuronal continuum cell death (Figure 2J), seen in all layers of the somatosensory cortex, were much less common than neuronal ischemic necrosis (Appendix A). HI-NT piglets had greater apoptosis (*p* = 0.006) in the anterior somatosensory cortex lateral bank than sham-NT piglets (Appendix A), but otherwise HI-NT and sham-NT piglets did not differ in apoptosis. In sham-HT piglets, apoptosis was suppressed compared to that in sham-NT piglets in the anterior somatosensory cortex lateral bank (*p* = 0.039) and mid-parietal cortex medial bank (*p* = 0.001) (Appendix A). The HI-HT group had greater apoptosis than the sham-HT group in the somatosensory cortex anterior medial (*p* = 0.03) and lateral (*p* = 0.01) banks and mid-parietal medial (*p* = 0.0005) and lateral (*p* = 0.01) banks. The HI-NT group had more (*p* = 0.01) apoptosis than the HT-HT group in the somatosensory cortex anterior lateral bank (Appendix A). Apoptosis–necrosis continuum cell death [66,69] (Figure 2J) was scarce in all groups except HI-NT piglets (Appendix A). It was most common in the medial bank of the mid-parietal somatosensory cortex in HI-NT piglets (Appendix A). In all somatosensory cortical areas, continuum cell death was significantly higher (*p* < 0.05) in HI-NT piglets than in HI-HT piglets.

#### 3.2.3. HT Strongly Protects the Primary Motor Cortex after HI

HI-NT piglets had a reduced normal-to-total neuron ratio in the motor cortex compared to sham-NT piglets (Appendix A). This effect varied by anteroposterior gyral level of the motor cortex (Appendix A). Normal neurons were significantly lost in HI-NT piglets at the anterior parietal medial (*p *= 0.001) and lateral (*p *= 0.001) banks and at the mid-parietal level medial (*p *= 0.004) and lateral (*p *= 0.001) banks of the motor cortex (Appendix A). HI-HT piglets had no statistical difference in the normal-to-total neuron ratio compared to sham-NT piglets (Appendix A). In contrast, HI-HT piglets, compared to HI-NT piglets, had significantly more normal neurons in the anterior parietal motor cortex medial (*p* = 0.01) and lateral (*p* = 0.003) banks and in the mid-parietal cortex medial (*p* = 0.01) and lateral (*p* = 0.002) banks (Appendix A). The medial prefrontal anterior-most motor cortex and gyrus rectus (Figure 2A) also showed loss of normal neurons in HI-NT piglets compared to sham-NT piglets (*p* = 0.006), but HI-HT piglets did not differ from HI-NT piglets (Appendix A). The lateral orbitofrontal cortex in HI-NT piglets also had significantly fewer normal neurons (*p* < 0.003) compared to sham-NT piglets (Appendix A), and this cortical region was also unprotected in HI-HT piglets.

HI-NT piglets had higher ratios of ischemic-necrotic-to-total neurons in the anterior and mid-parietal motor cortex compared to sham-NT and HI-HT piglets (Figure 8). Specifically, in the anterior parietal motor cortex in HI-NT piglets (Figure 2B), the medial gyral (*p *= 0.004) and lateral (*p *= 0.001) banks had more ischemic-necrotic neurons compared to sham-NT piglets (Figure 8A,B). The mid-parietal motor cortex (Figure 2B) showed a similar pattern, with more ischemic necrosis in the medial (*p *= 0.003) and lateral (*p* < 0.001) banks (Figure 8C,D) in a HT-NT to sham-NT comparison. In the anterior-most region of the motor cortex located in the medial prefrontal cortex (Figure 2A), HI-NT piglets had increased (*p* = 0.008) ischemic-necrotic neurons compared to sham-NT piglets (Figure 8E). The lateral orbitofrontal cortex in HI-NT piglets also had a higher ratio of ischemic-necrotic-to-total neurons compared to sham-NT piglets (Figure 8F). Interestingly, the frontal cortex seemed less vulnerable to ischemic neuron degeneration than the more posterior neocortex (Figure 8A–F). Moreover, the increased motor cortical ischemic-necrotic-to-total neuron ratios of the anterior and mid-parietal gyri in HI-NT piglets were all significantly diminished in HI-HT piglets, but prefrontal cortex vulnerability was unaffected by HT (Figure 8A–F).

Apoptotic profiles (Figure 2K) were assessed in the motor cortex (Appendix A). Apoptotic cell counts in HI-NT and HI-HT piglets were not significantly different from those in both sham groups (Appendix A). However, apoptosis was significantly suppressed in sham-HT piglets compared to sham-NT piglets (Appendix A) in the anterior parietal gyrus medial (*p* = 0.03) and lateral (*p* = 0.01) banks, middle parietal cortex medial bank (*p* = 0.001), and frontal cortex medial (*p* = 0.02) and orbitofrontal (*p* = 0.04) gyri.

Continuum cell death profiles (Figure 2J) were assessed in the motor cortex (Appendix A). Although the cell death continuum manifested more often in HI-NT piglets than in sham-NT piglets, particularly in the lateral bank of the mid-parietal cortex (Appendix A), no significant differences were detected.

To examine the potential effects of anesthesia with and without HT, we compared the ratio of normal-to-total neurons among naïve unanesthetized (*n* = 6), sham-NT (*n* = 6), and sham-HT (*n* = 10) piglets. The ratio of normal-to-total neurons did not differ between the anterior and mid-parietal somatosensory and motor cortices among the unanesthetized and sham piglets (*p* > 0.05 for all comparisons).

### 3.3. Neuropathology in Neocortex of HI Piglets Involves Laminar Vulnerability and Depletion of Nuclear Rbfox3

Because our other neuropathological assessments in HI piglets (Figure 7 and Figure 8; Appendix A) were conducted independent of cortical layer, we examined the inferior parietal and somatosensory cortices more closely for specific laminar pathology. The piglet neocortex has very discernible layers (Figure 3A,B, Figure 4B–D, Figure 5C,J and Figure 7E; Appendix A). The inferior parietal gyrus is easily identifiable for characterization in piglets (Figure 2B,C). This region was interrogated for neuropathology by H&E staining (Figure 9) because it appeared to be damaged in seizing piglets but not as a result of primary HI injury [19,20]. Layer III neurons in a sham-NT piglet are illustrated in Figure 9A. All neuron profiles present were normal, and the neuropil was intact without perineuronal swelling or spongiform encephalopathy (Figure 9A). In HI-NT piglets, layer III of the inferior parietal cortex was severely depleted of normal neurons (Figure 9B,C) with only ischemic-necrotic neuronal tombstones remaining and extensive encephalopathic spongiform changes in the neuropil (Figure 9B), similar to the pattern shown for the somatosensory cortex (Figure 2I). Layer III was protected in HI-HT piglets compared to HI-NT piglets (Figure 9C).

We used IHC for Rbfox3 as a neuronal marker to further verify the identities of somatosensory and motor cortices in piglets (Figure 3A,B; Appendix A) and to reveal potential aberrancies in its neocortical neuron localization after HI. The NeuN antibody [85] used to detect Rbfox3 in the piglet brain was specific, as shown by Western blotting after SDS-PAGE wherein expected proteins with mobilities at 46 and 48 kDa were seen (Figure 3A inset; Appendix A). Total neuron counting was performed using the Nissl counterstain. In the sham piglet neocortex, Rbfox3 defined all six layers with particular discernment, depending on known functional localizations (Figure 3A,B; Appendix A). In the somatosensory cortex, layer IV was prominently thick (Figure 3A). In contrast, the motor cortex had an attenuated layer IV and prominent layers V and VI (Figure 3B). Rbfox3 had nuclear and cytoplasmic localization (Figure 3B, inset, E). Within the nucleus, Rbfox3 was present in numerous subnuclear particles (Figure 3B,E).

Rbfox3 was then used as a marker to count neurons (Figure 10A–C). In HI-NT piglets compared to sham-NT piglets (Figure 10), significant loss of total neurons was detected in layers II (*p* < 0.0001) and III (*p* < 0.0001), while layer V had significant loss (*p* < 0.0005) but less vulnerability than layers II and III. In HI-HT piglets compared to HI-NT piglets, neuronal densities were partially rescued in all layers (Figure 10A–C).

In HI piglets, Rbfox3 positivity in the somatosensory cortex discriminated additional patterns of injury unseen by H&E staining. Layer II neurons in the somatosensory cortex were depleted of Rbfox3, while in the same tissue section, layer II neurons in the adjacent motor cortex were positive (Figure 3C,D). In the somatosensory cortex, layer II, III, and V neurons were characterized by their Rbfox3 staining patterns. Rbfox3 immunoreactivity was affected by HI and HT treatments (Figure 10D–F). Sham-HT piglets compared to sham-NT piglets had significantly higher Rbfox3-depleted-to-total neuron ratios in layers II (*p* = 0.002), III (*p* = 0.01), and V (*p* = 0.001). HI-NT piglets compared to sham-NT piglets (Figure 10D–F) had significantly higher Rbfox3-depleted-to-total neuron ratios in layers II (*p* < 0.0001), III (*p* < 0.0001), and V (*p* < 0.0001). The increased Rbfox3-depleted-to-total neuron ratio seen in HI-NT piglets was mitigated in HI-HT piglets in layer V (Figure 10D–F). Interestingly RBfox3-depleted neurons generally appeared morphologically normal based on Nissl counterstaining and residual Rbfox3 immunoreactivity (Figure 3E). Subsets of neurons degenerating as ischemic-necrotic or continuum forms were strongly positive for Rbfox3, with the immunoreactivity favorably delineating the nuclear clumping of continuum degeneration (Figure 3C inset).

### 3.4. HT Reduces Neuronal Injury in the Putamen after HI

Neuronal damage quantified in the putamen generally corresponded to axonal target areas of the motor and somatosensory cortices (Figure 5F,L). Compared to sham-NT piglets, HI-NT piglets had a highly significant decrease (*p* < 0.0001) in the ratio of normal-to-total neurons (Appendix A) and a highly significant increase (*p* < 0.0001) in the ratio of ischemic-necrotic-to-total neurons (Appendix A). Compared to the HI-NT piglet putamen, the HI-HT piglet putamen had significantly more (*p* < 0.001) normal neurons and significantly less (*p* < 0.001) ischemic-necrotic neurons. The number of apoptotic cells did not differ among treatments (Appendix A), but continuum cell death was increased (*p* = 0.049) in HI-NT piglets compared to that in sham-NT piglets (Appendix A).

### 3.5. Electrographic and Clinical Seizures

Eighteen piglets underwent cEEG and video monitoring. The cEEG-determined seizure burden was not affected by temperature in the two small cohorts of HI piglets. Seizure burden was significantly higher in HI-NT piglets than in sham-NT piglets (*p *= 0.05; Figure 11A). Normal piglet cEEG patterns showed activity of various frequencies, primarily high frequency (>10 Hz) and voltages of 0.1–0.2 mV, with variations in frequency and amplitude throughout all leads (Figure 11B). A few sham piglets (these piglets were anesthetized for ~30 h) showed a low level of electroencephalographic seizure activity. Clinical seizures usually emerged within 24 h after extubation during the second day after HI. They often appeared during sleep, consisting of orofacial twitching, tongue movements (Appendix A), rooting movements of the snout and head, repeated jerks of the head and legs, sudden arousal, and then clonic movements (see Appendix A). The clonic movements spread from the head region to the shoulder and forelegs to form generalized seizures (see Appendix A). Some piglets developed fictive running movements when the seizure was believed to progress to status epilepticus (see Appendix A). cEEG confirmed the presence of seizures with rhythmic spike-wave complexes of higher voltage (>0.4 mV) in specific neocortical areas that generalized to other areas of the neocortex (Figure 11C, Appendix A). Some cEEG-detected seizures appeared subclinical or equivocally clinical (Appendix A).

Greater seizure burden was correlated with more neuronal ischemic necrosis (panlaminar) in the somatosensory cortex at the anterior parietal (medial bank: r = 0.57; *p *= 0.015; *n* = 18) and mid-parietal (medial: r = 0.69; *p *= 0.002; lateral: r = 0.62; *p *= 0.006) levels (Figure 12). Ischemic necrosis in the orbitofrontal cortex was also correlated with seizure burden (r = 0.48; *p* = 0.045). Additionally, more ischemic necrosis in the putamen was correlated with higher seizure burden (r = 0.60, *p *= 0.009; Figure 12E). The seizure burden did not correlate with the amount of ischemic necrosis in other cortical areas (*p* > 0.05 for all comparisons). Normal neuron number in layer III of the inferior parietal cortex did not correlate with seizure burden in this small sample size (Appendix A).

## 4. Discussion

This work is a novel contribution to experimental neonatal brain neuroanatomy, pathophysiology, and neuropathology. We show that piglets are a translationally relevant, neonatal, large non-primate animal model to study the cytoarchitecture, connectomics, and electroencephalography of the immature gyrencephalic cerebral cortex. In parallel, we developed a new survival model of global HI with cEEG monitoring during recovery from HT to study neurological phenotypes and neocortical neuropathology in neonatal piglets. We answered several questions as outlined in the Introduction: (1) The neonatal piglet neocortex has differential spatial vulnerability to HI (Figure 7, Figure 8 and Figure 9, Appendix A) seen in the somatosensory, motor, inferior parietal, and prefrontal cortices that is differentially protected by HT. (2) cEEG-confirmed seizures are associated with greater ischemic-necrotic neurodegeneration and neuron loss after HI (Figure 12). (3) HT initiated 2 h after HI robustly protects, although variably, against neuropathology in functionally different regions of the neonatal gyrencephalic neocortex (Figure 7, Figure 8 and Figure 9, Appendix A) but does not protect against electroencephalographic seizures (Figure 11, Appendix A). (4) Panlaminar ischemic-necrotic neurodegeneration in the somatosensory and orbitofrontal cortices is seizure-related encephalopathy (Figure 7 and Figure 12). (5) As shown in H&E staining, neurons in the neonatal neocortex of HI piglets have a limited repertoire in their morphologic degeneration (ischemic necrosis is the predominant form) in the presence or absence of seizures (Figure 2 and Figure 7). However, nuclear depletion of cortical neuronal RNA splicing protein Rbfox3 (Figure 3 and Figure 10) is a novel immunophenotype of neonatal HI brain damage with seizures, and it could be a molecular mechanism involved in seizure origin.

### 4.1. Animal Models to Study Neocortical Organization, HI Injury, and Seizures in the Neonatal Brain

An important aspect of experimental work is its translational applicability to human disease and injury. In human neonatal HIE, the peri-Rolandic area of the neocortex is often damaged [10,11,12,14,86], meaning the frontal lobe-located primary motor cortex in the anterior bank of the central sulcus of Rolando and the parietal lobe-located primary somatosensory cortex in the posterior bank and bottom of this sulcus. These regions are contiguous as Brodmann area 4 merges with Brodmann area 3a. The ansate sulcus in the piglet, not present in rodents because of their lissencephaly [87], is homologous to the human central sulcus [88]. In neonatal piglets, we cytologically identified the agranular motor cortex by its minimized layer IV, prominent layer V, large Betz pyramidal neurons, and its projections to the central putamen. Similarly, we identified the somatosensory cortex by cytology and its connectivity with the putamen, thalamus, and cortico-cortical connections. The cytology was similar to what has been described for humans [77,84,89,90]. We evoked limited motor responses and somatosensory-evoked potential recordings to assist in identifying these regions, which corresponded to the electrophysiological mapping of pigs conducted by other researchers [82,91].

Human infants with HIE plus seizures are likely to have an abnormal brain MRI and persistent neurodevelopmental impairments despite having received HT [92]; therefore, a better understanding of the neuropathology and mechanisms of neonatal seizures and long-lasting functional deficits ensuing from neonatal HI brain injury in translationally relevant animal models can help formulate therapeutics adjuvant to HT. Animal models will drive the understanding of relationships between neonatal HI brain injury, seizure co-morbidity, neuropathology, and impairments in cognitive, motor, and affective functioning. In mice and rats, seizures early in life can lead to cognitive, social, and behavioral disorders [93] with changes in hippocampal synaptic plasticity and memory [94]; however, rodents are lissencephalic [87] and have very different brain development chronicity and primary sensory-driven, including binocularity of vision, social structure, compared to humans. To best model human neonates, a time-equivalent gyrencephalic species with a structurally similar six-layered neocortex is desirable [95]. Term fetal rhesus monkeys subjected to partial asphyxia have seizures that aggravate the neuropathology [33]. Fetal sheep have been used as HI brain injury models to show therapeutic efficacy of HT [96] and that status epilepticus following severe HI is associated with greater brain injury [97]. However, generally sheep models have been short survival without extended monitoring for neurological outcomes. Ideally, it is best for injured animals to survive with an independent upright and mobile status for feeding and observer assessments of posture, muscle tone, balance, gait, and cognition. At a cellular level, neocortical pyramidal neurons and interneurons in animals should have archetypal biology like human neurons. For example, although gyrencephalic, ferrets have 5-layer pyramidal neurons with very different biophysical properties compared to human neurons, which can affect excitability and excitotoxicity [98]. We have shown that pig cortical neurons derived from directed differentiation of neural stem cells [74] have, compared to human neurons, more similar responses to excitotoxins and oxidative stress than mouse cortical neurons [18].

Therefore, we sought to develop a model that comprehensively integrates clinically relevant HI brain damage in a large gyrencephalic animal, a therapeutic standard of clinical care (HT), survival, cEEG, neurologic phenotypes, and cell-resolution neocortical neuropathology. The size of the neonatal piglet cranium and brain gyrencephaly allows precise transcranial electrode placement to monitor EEG in specific gyri in the immediate post-HI and HT periods for many days. This contrasts with cEEG monitoring in rodents that begins long after HI [99] because newborn pups cannot be separated from a dam for extended periods for EEG monitoring [100,101]. Because of longstanding ethical and moral concerns about using nonhuman primates in research [102], particularly infant rhesus monkeys [103,104], our work here and other work [105] demonstrates that the piglet is a realistic, clinicopathophysiologically relevant, and legitimate neuroanatomical and neuropathological alternative for survival modeling of neonatal HI brain injury with the standard of care, co-morbid seizures, and long-term neurological outcomes.

The ability to continuously monitor post-HI EEG in neonatal piglets provides a key opportunity to study the molecular and cellular interactions between HI, seizures, and secondary brain injury with a physiologic read-out for testing adjuvant treatments to HT. This effort is important because the clinical management of neonatal seizures is debated [23]. Nearly half of neonatal seizures from HIE are subclinical [106]. A randomized clinical trial of neonates cooled for HIE showed no difference in death or 2-year disability, including blindness, deafness, deficits in cognition and language, as well as motor, social-emotional, and adaptive skills between children who did or did not receive anti-seizure medication (phenobarbital or phenytoin) for electrographic or clinical seizures [106]. However, site discrepancies in seizure management caused study underpowering and premature ending; thus, uncertainty remains in the predictive relationship between treating neonatal seizures and persistent childhood disability [106].

We found that the timing for general tonic-clonic clinical seizures in piglets corresponded with seizures in neonatal humans and monkeys, which typically begin within 1–2 days after birth injury [32,107,108]. Video recordings confirmed seizure activity, including general tonic-clonic seizures in some piglets (see Appendix A). Among a subset of piglets monitored by EEG, HI increased the electroencephalographic seizure burden. Although some piglets needed to be euthanized prior to the endpoint survival of 7 days because of unresolving seizures, animals surviving to 7 days and beyond could be subjected to future neurological examination and cognitive testing. Cognitive status assessment using a spatial T-maze is feasible to test for deficits that may be related to HI brain injury and seizures. We previously found that cognitive deficits in the T-maze are associated with hippocampal CA1 interneuron attrition and CA3 neuron loss in piglets with HI, although clinical seizures were not seen in this non-cooled cohort [105].

### 4.2. Our Piglet Pathophysiology Mimics Clinical Neonatal HIE

Our hypoxia-asphyxia, CPR, and therapeutic HT protocol is clinically relevant. The hypoxia-asphyxia episode caused severe hypotension with respiratory and metabolic acidosis. PaCO_2_ levels after resuscitation aligned with clinical observations [61,109]. The target temperatures during HT and rewarming were strictly monitored and achievable. Hemoglobin levels were higher during HT in HI piglets and then showed no differences once rewarming began. This is consistent with expected increases in measured hematocrit during HT [110,111]. Stress-induced hyperglycemia occurred immediately after resuscitation from HI, as expected with severe injury [112]. Subsequent glucose levels tended to be higher in the HT groups. Early hyperglycemia is associated with greater brain injury on MRI and poorer neurodevelopmental outcomes [112]. This early hyperglycemia is likely related to the sympathoadrenal response engaged during asphyxia and was observed in both HI groups. Hyperglycemia was also seen at the end of cooling in HI and sham piglets, suggesting sustained sympathetic activity or decreased tissue glucose utilization. However, we did not administer insulin in our protocols for safety reasons because hypoglycemia is also associated with poor neurologic outcomes, cardiovascular instability, and seizures [112,113]. Instead, the glucose content in IV fluids was adjusted to reduce the incidence of hyperglycemia.

### 4.3. The Neonatal Piglet Neocortex Has Suprasylvian and Prefrontal Vulnerability to HI

Many children surviving neonatal HIE and HT have, compared to peers without history of injury, IQ deficits and moderate-to-severe impairments in attention-executive, visuospatial, and emotional-social functions, and language skills [4,6]. The neocortex drives these functions [7], and the prefrontal cortex reigns supreme [8,9]; thus, these persisting childhood deficits suggest that: (1) the neocortex has regional topographic vulnerability in neonatal HIE; (2) vulnerable neocortical areas are insensitive or less responsive to HT compared to deep gray matter; and (3) damage to the neocortex can develop after the injury-cooling period perhaps with delayed, remote, and trans-synaptic degenerative features. These possibilities are consistent with clinical observations and with observations in HI rodents [114,115]. Term HIE infants show selective neocortical highlighting on T1-MRI involving the superior peri-Rolandic, posterior cingulate, and insular cortex [14]. These abnormal signals can take several days to manifest and then peak during the second week after insult [14]. The peri-Rolandic watershed damage in HIE is correlated with verbal IQ at 4 years of age [116]. Although gyrencephalic, the pig brain is not furnished with a Rolandic central sulcus or extensive Sylvian lateral fissure; however, there is a Sylvian sulcus and superior to this is the primary somatosensory cortex and posterior motor cortex homologous to the human peri-Rolandic cortex [88]. Their vulnerability to neonatal HI was shown here and previously [19,20]. These regions were protected after HI by overnight HT. We also examined, for the first time, the prefrontal anterior motor cortex and inferior orbitofrontal cortex in our piglet model. Interestingly, in contrast to the high vulnerability of the posterior regions of the motor cortex, the anterior motor cortex was less vulnerable to HI. The cortical area assessed here also included the gyrus rectus, which corresponds to Brodmann area 11 and functions in decision making in context of reward [117]. The lateral inferior prefrontal cortex was damaged by HI and protected by HT. This area roughly corresponds to Brodmann areas 10 and 47 in humans [81]; these cortical regions have executive and cognitive flexibility functions [118,119]. Close examination of these areas (Figure 2A) with MRI in clinical neonatal HIE could yield new information relevant to the functional deficits seen in school-aged children with early life HIE and cooling.

We also quantitatively examined, for the first time, the inferior parietal cortex in our piglet model. Piglets with seizures had damage in this area consistent with previous observations based on clinical seizures in non-cooled HI piglets [19,20,36]. This neocortical region may correspond to the vestibular/auditory cortex identified in monkeys, cats, and humans [82,120], allowing for the possibility that damage to this region is related to the emergence of generalized tonic-clonic seizures and decerebrate posturing through release of the lateral vestibular nucleus and activation of the lateral vestibulospinal pathway that strongly excites antigravity extensors [121,122,123]. Lateral vestibular nucleus neurons (Deiter’s neurons) forming the vestibulospinal tract are tonically active [123]. Like the descending influences of suprasegmental pathways on the spinal cord through inhibitory interneurons [121,124], the cerebral cortex projects to the lateral vestibular nucleus mediating inhibitory modulation [125]. This pathway, alone or in combination with damage to the cerebellum, another potent inhibitory modulator of Deiter’s neurons [122,123], could be inactivated in HI piglets with neocortical damage. “Released” central pattern generators in the brainstem [126,127] could also be involved in the fictive locomotor activity we observed and documented by video recording in some piglets with seizures (see Appendix A).

We specifically performed a laminar analysis of layers II, III, and V because layer-specific cortico-cortical connectivity can mediate seizure propagation [128], and layers II and III have emergent synaptic mechanisms controlling pyramidal neurons in the gyrencephalic brain that are human-specific [95,129]. Moreover, layer III frontal cortical neurons show dendritic spine depletion in human epilepsy [130]. Layers II, III, and V showed very significant loss of neurons in HI-NT piglets, consistent with our previous work [19]. Layers II and III appeared to be more vulnerable than layer V. Previous work in a rat model of epilepsy showed that layer III is preferentially vulnerable [131]. A recent RNAseq study of human epilepsy identified layer II–III Cux1-positive principal neurons and layer V Fexf2-positive neurons as possible drivers of excitability perturbations in the neocortex [132].

Seizures appeared unmitigated by HT in piglets after HI in this study with small group sizes. HT was initiated 2 h after HI to approximate delays encountered in hospital practice. Our HT protocol was for 29 h, thus contrasting with the longer HT periods used clinically [60,61] and in fetal sheep [96,97]. In HI piglets, HT for 29 h was sufficient to provide robust neocortical neuron protection, as shown here, and sustained neuroprotection in the putamen [67]. Other work in piglets suggested that overnight HT could prevent clinical seizures, but lesser insult and slightly older animals were used [133]. Clinically, 40–70% of neonates cooled for HIE have seizures [22,94]. Clinical seizures and abnormal EEG in human neonates are associated with injury in the neocortex and basal ganglia [94,108]. We also found that greater seizure burden was correlated with more panlaminar ischemic necrosis in the anterior and mid-parietal somatosensory cortex, frontal motor cortex, inferior parietal cortex, and in the putamen. HT reduced this panlaminar ischemic-necrotic neuron number. Thus, a dichotomy emerges with HT affecting the burden of neuronal cell body damage but not seizure burden. Perhaps seizures are independent of the primary HI brain damage that is responsive to HT. However, these HT-rescued neuronal cell bodies may be functional, non-functional, or aberrantly functional. Preservation of a particular domain of neurons does not necessarily mean preservation of appropriate function. The cellular and molecular pathobiology of the inability of HT to protect against seizures needs identification. Perhaps there is a relationship with HT duration or Rbfox3.

### 4.4. Neocortical Neuronal Degeneration in Piglets after HI and HT Has a Limited Structural Phenotype Repertoire

This study is our first detailed multiregional examination of neurodegeneration in the piglet neocortex after HI, overnight HT, and co-morbid seizures. In addition to determining whether there is differential regional and laminar vulnerability of the neocortex in HI piglets with and without cooling and whether cooling protects the neocortex, we examined neuronal cell death phenotypes. Classic ischemic-necrotic neurodegeneration was the predominant phenotype in the neocortex independent of temperature and seizure presence. More ischemic necrosis in the anterior and mid-parietal somatosensory cortex, lateral orbitofrontal cortex, and putamen was correlated with higher seizure burden. Importantly, HT and seizure presence did not alter the neurodegeneration morphology seen by H&E staining, and, although we detected apoptotic and continuum forms of cell degeneration in the piglet neocortex, the predominance of these phenotypes was modestly affected by treatment.

An ischemic-necrotic neuron majority in the cerebral cortex is consistent with our observations in the piglet striatum [56,73,133]. Critically, HI and HT as well as co-morbid seizures did not induce neuronal cell death forms other than eosinophilic necrosis in piglets. One expectation was that cooling slows the kinetics of neuronal injury and shifts the ischemic necrosis to a hybrid continuum or apoptotic form of cell death [134] (see Figure 2). We observed neuronal cell death phenotype shifting in other forms of degeneration in mouse systems [135,136,137]. This was not the case in the piglet neocortex within our 2 to 7 day survival period. An immutable form of neuronal cell death also appears to be the case in the putamen, wherein we saw protection with HT. Cell death phenotype switching was also absent in our previous studies of HT protection of the putamen with piglet survival of 3 and 6 h [68] and 1 and 10 days [67], although in these cohorts with less severe insult, clinical seizures were not observed. These observations are meaningful because of their relevance to adjuvant therapy to HT. If this holds true for human neonatal HIE and cooling, although it might not because of human and non-primate neuron species differences in cell death mechanisms [18], then regardless of targeted temperature management and the presence or absence of seizures after HI, subacute therapies targeting cell death will likely need to modulate cellular necrosis.

### 4.5. Rbfox3 Immunophenotyping Identifies a New Form of Neocortical Neuropathology in Neonatal HI Brain Injury

The neuron-specific nuclear protein NeuN is Rbfox3, a member of the RNA-binding protein Rbfox-1 gene family [138]. Broadly, RNA-binding proteins have been critical for the evolution of neocortical expansion and transition from lissencephaly to gyrencephaly [139]. Rbfox3 is found in subnuclear structures called speckles (interchromatin clusters); it shuttles between the nucleoplasm and nuclear matrices [140]. Rbfox3 functions in mRNA splicing and cryptic exon suppression [141]. Rbfox family members target several candidate genes associated with epilepsy that encode glutamic acid decarboxylase, GABA receptors, glutamate transporters, potassium channels, and sodium channels [38,142]. Mutations in the *Rbfox3* gene have been identified in patients with neonatal epilepsy [38,44,46], developmental delay, and language/speech disorders [143,144]. Knockout mice have shown that Rbfox3 is required for maintenance of the balance of excitatory-inhibitory circuitry [145] by regulating vesicle-associated membrane protein 1 expression [146]. Another downstream function of Rbfox3 protein is assembly of the axon initial segment, which is a key axonal microdomain involved in neuronal excitability [147].

We used a monoclonal antibody that has been well characterized by others [85,148] and us [77,78] (Figure 3A inset) to detect Rbfox3 in piglet brain sections. In normal piglets, virtually all neocortical neurons had nuclear positivity for Rbfox3 in the somatosensory, motor, and inferior parietal cortices. As in other species [148], piglet neocortical neurons had nuclear and cytoplasmic Rbfox3 immunoreactivity (Figure 3B,F, inset; Appendix A). Rbfox proteins in the cytoplasm regulate gene expression by binding to 3′-untranslated regions in target mRNA [149]. Rbfox proteins in the nucleus bind to intronic consensus UGCAUG sequences that flank alternate exons to regulate pre-mRNA splicing [150]. We identified three types of neurons in the sham and HI piglet neocortex: Rbfox3-positive/normal, Rbfox3-positive/ischemic-necrotic or continuum degeneration, and Rbfox3-depleted neurons. Degenerating neurons in the neocortex were positive for Rbfox3 but had mislocalized and aggregated Rbfox3. Rbfox3-depleted neurons appeared in layers II, III, and V in HI piglets. The reason for the loss of Rbfox3 immunopositivity in otherwise normal appearing neurons and its consequences are unclear. The NeuN antibody epitope maps to an N-terminal 16 amino acid sequence with a tyrosine-proline residue pair being critical for binding [148]. With tyrosine being part of the epitope, antibody binding could be phosphorylation dependent. Depletion of Rbfox3 staining in subsets of neurons that appear normal (not ischemic-necrotic, apoptotic, or continuum) by Nissl staining in the piglet somatosensory cortex could be biochemically and functionally tantamount to somatic de novo loss-of-function mutations in the *Rbfox3* gene in neocortical neuron ensembles with altered excitability and function caused by exonic “poisons,” as in Dravet syndrome [151]. If this turns out to be the case, and because *Rbfox3* mutations are a known cause of epilepsy in infants [44,46], then this Rbfox3 nuclear depletion phenotype could illuminate a molecular mechanism for seizures or cryptic exon neuropathology in neonatal HI brain injury. The nuclear restoration of Rbfox3 with HT could drive neuroprotection, although it might be a signature of neuronal rescue unrelated to seizures. Our Rbfox3 observations in the primary somatosensory cortex are also pertinent because of the propinquity to the perisylvian epileptic network proposed in humans [152]. The biology of Rbfox3/NeuN is extremely complex and dynamic [141], with negative immunoreactivity divergent from neuronal loss or degeneration [153]. In future experiments, we will perform RNAseq to examine RNA splicing-regulated nonsense-mediated decay of epilepsy-related gene targets and axon initial segment integrity of the neocortical neuron subtypes in our piglets.

### 4.6. Study Caveats and Future Directions

This work has noteworthy experimental design and interpretational limitations. The treatment group sizes are small because of the intensiveness of the model. Only male piglets were studied because we have not seen sex-biased neuropathological outcome differences among male and female neonatal piglets (Martin LJ, Lee JK, unpublished observations) and anticipate that the results described here apply to female piglets. HT was not performed for 72 h like the clinical protocol [60,61], so ineffective protection against seizures could be related to the duration of HT. Studies in fetal sheep show that cooling for 72 h seems ideal [154]; however, we used continuous anesthesia during HT in piglets, so to avoid prolonged anesthetic exposure and possible collateral pathophysiology, the HT duration was shortened. The H&E neuropathology assessment was limited, being based only on the neuronal cell body and entirely exclusive of damage to dendrites, axons, and other components in the neuropil compartment that could be critical to seizure initiation and propagation. Piglet seizures were allowed to self-resolve without pharmacologic treatment, and piglets with continuous clinical seizures were euthanized. We did not confirm the clinical and cEEG-defined seizures by blocking them with known anti-epileptics. Without seizure interventions, we could not determine unequivocally whether the seizures induced secondary brain injury and propagated additional seizures or if the seizures strictly resulted from the primary HI injury. Lastly, we did not identify the seizure mechanisms at the cellular, connectome, or molecular level, although we gleaned evidence that cellular necrosis, layers II–IV, and Rbfox3 might be involved. The interface between lower layer III and upper layer IV particularly interests us. In the future, the neurologic and neuropathologic (cellular and connectome-wide) outcomes of treating post-HI seizures with different anti-seizure medications will be assessed and layer-specific bulk RNAseq will be performed in piglets with and without HT.

## 5. Conclusions

We conclude that: (1) the neonatal piglet neocortex has suprasylvian vulnerability to HI and seizures; (2) neonatal HI piglet neuron cytopathology repertoire is limited, and HT protects functionally different regions of the neonatal neocortex; (3) higher seizure burden is correlated with more panlaminar ischemic-necrotic neurons in the somatosensory cortex; (4) seizures appear to be insensitive to HT; and (5) HI cortical neurons have an RNA splicing protein nuclear depletion. This work demonstrates with a gyrencephalic large animal survival model of neonatal HI brain damage, mild HT and rewarming, and cEEG monitoring that the neocortex has topographic and laminar vulnerability possibly influenced by seizure burden and RNA splicing factor pathology.

## Figures and Tables

**Figure 1 cells-12-02454-f001:**
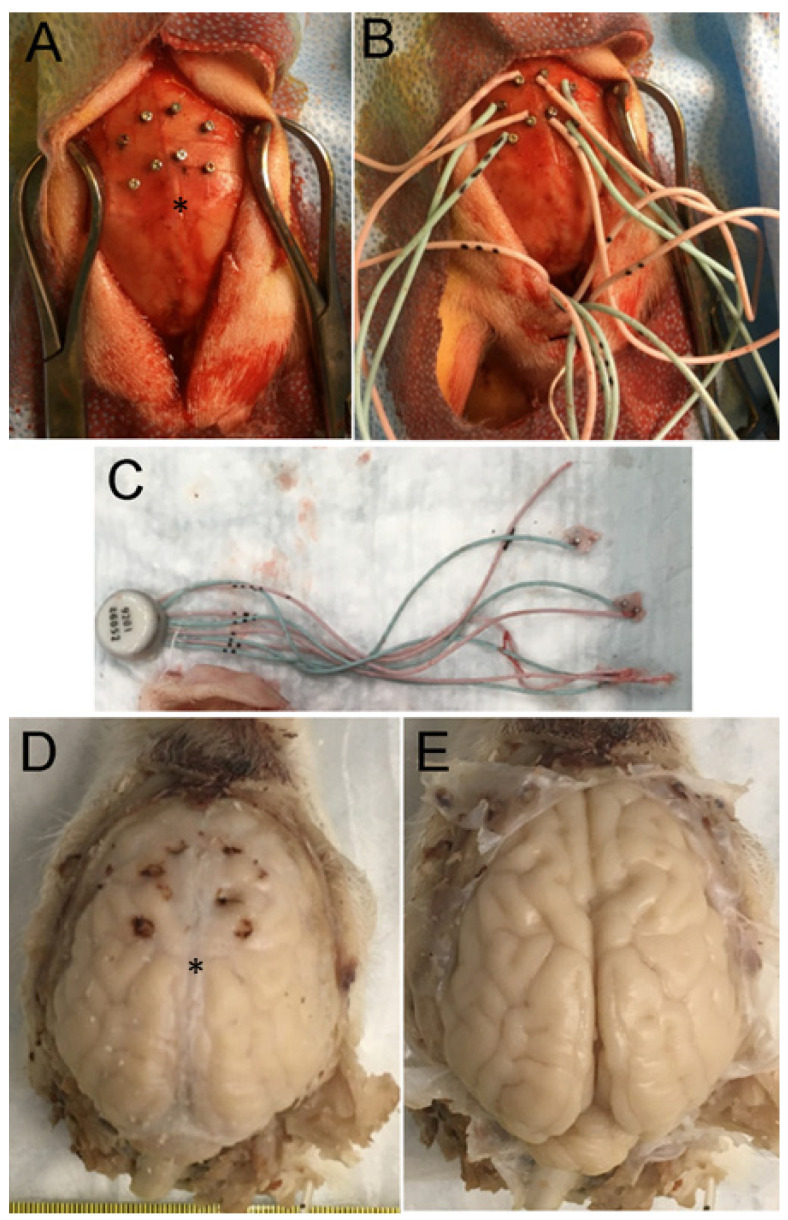
Surgical placement of electrodes for cEEG in neonatal piglets. (**A**) Cranial screw placement. (**B**) EEG electrode array attachment. (**C**) Isolated EEG electrode array showing the round transmitter (on left) and naked electrodes and electrodes with attached cranial screws and acrylic cement placodes. (**D**) Exposed brain of a 4% PF perfusion fixed piglet with the calvarium removed and intact dura mater, showing the dural placement sites of cranial screws seen as focal hemorrhagic damage. (**E**) The same brain shown in (**D**) with the dura mater removed and no damage to the underlying neocortical parenchyma. The optimal quality of the brain perfusion fixation is also demonstrated by the absence of residual blood in cortical vessels and discrete contouring of the cortical surface. Asterisks in (**A**,**D**) identify bregma.

**Figure 2 cells-12-02454-f002:**
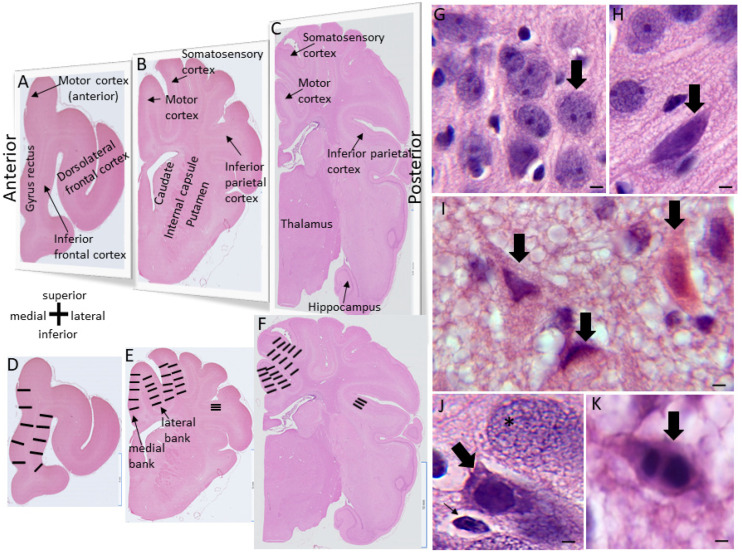
Piglet brain used for neuron counting in three neuroanatomic levels, H&E counting strategies, and neuronal injury classification. (**A**–**C**) H&E-stained hemibrain sections at the levels of frontal cortex (**A**), posterior frontal-anterior parietal cortex and striatum (**B**), mid-parietal cortex, posterior striatum, and thalamus (**C**). Brain regions pertinent to the study are identified. Medial is at left. Superior is at top **(see +)**. (**D**–**F**) Neurons were counted in rows of adjacent microscopic fields in cortical layers II–VI or in specific layers in the inferior parietal cortex. In the frontal cortex (**A**,**D**), counts were performed in the medial-most gyrus comprised dorsally of the anterior motor cortex, the ventrally located gyrus rectus, and laterally of the inferior frontal (orbitofrontal) cortex. At a striatal level (**B**,**E**), counts were performed in the medial and lateral banks of the motor cortex and somatosensory cortex and in layers II, III, and V of the inferior parietal cortex (horizontal black bars). At a thalamic level (**C**,**F**), counts were performed in the medial and lateral banks of the posterior motor and somatosensory cortices. (**G**–**K**) Neuron morphology classification by H&E staining (see methods for descriptions): normal neuron ((**G**), arrow), injured neuron ((**H**), arrow), ischemic-necrotic degenerating neurons ((**I**), arrows), apoptosis-necrosis continuum degenerating neurons ((**J**), arrows), and apoptotic neuron ((**K**), arrow). In panel (**J**), neurons are identified as normal (asterisk), early continuum cell degeneration (large broad arrow) with nascent irregular chromatin clumping in the nucleus, and near end-stage continuum degeneration (small thin arrow) with severe neuron attrition and non-apoptotic nuclear condensation. Note the optimal brain perfusion fixation in all images. Scale bars (in µm): (**G**), 10; (**H**), 10; (**I**), 8.75; (**J**), 10; (**K**), 25.

**Figure 3 cells-12-02454-f003:**
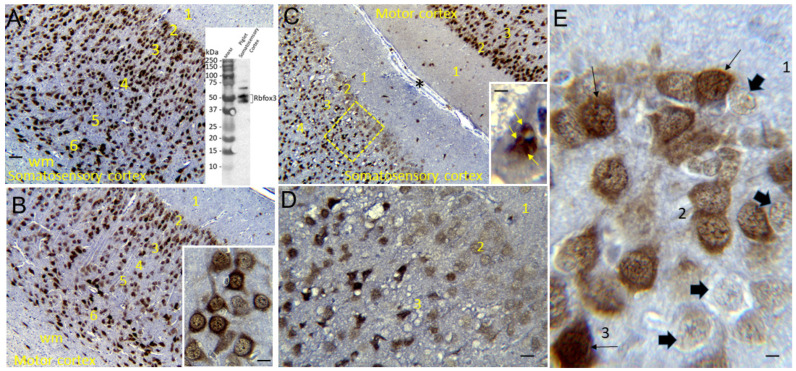
Rbfox3 detection in the piglet neocortex with NeuN antibody and immunoperoxidase immunohistochemistry. (**A**) In the sham piglet somatosensory cortex, RbFox3-positive cells have a laminar distribution (brown immunoreactivity with blue Nissl counterstaining). Numbers identify the cortical layers. A thick layer 4 is discernable. Inset shows antibody specificity in Western blot of crude homogenates of the piglet somatosensory cortex with prominent band detection at 46–48 kDa. (**B**) With a prominent layer 5 and attenuated layer 4, RbFox3 immunostaining in the sham piglet motor cortex is distinct from that seen in the somatosensory cortex. Inset shows the nuclear and cytoplasmic staining for Rbfox3 in piglet neocortical neurons. (**C**) In HI-NT piglets, Rbfox3 is selectively depleted in layer II neurons of the somatosensory cortex, yet across the sulcus (asterisk), motor cortex layer II neurons are positive. Other pyramidal neurons (inset) in layers 3-5 have nuclear clumping of Rbfox3 when undergoing continuum degeneration based on large irregular nuclear aggregates. Hatched box is shown at higher magnification in (**D**). (**D**) HI-NT piglet with selective depletion of Rbfox3 in layer 2, yet the neurons appear to be morphologically normal. (**E**) Somatosensory cortex layer II of an HI-HT piglet with depletion of Rbfox3 in some neurons (solid broad arrow) and partial rescue of Rbfox3 positivity in other neurons (solid thin arrows). Scale bars (in µm): (**A**) (same for (**B**,**C**)), 160; (**B**) inset, 9; (**C**) inset, 10; (**D**), 18; (**E**), 7. See Appendix A for additional data.

**Figure 4 cells-12-02454-f004:**
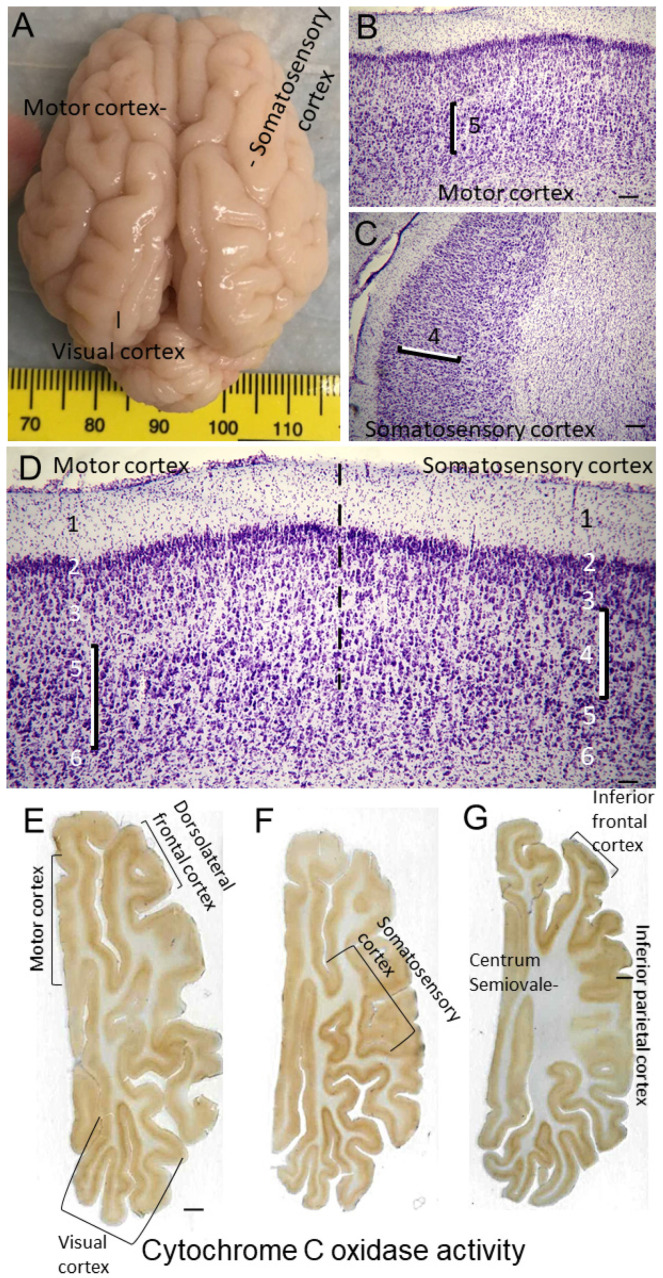
Neonatal piglet neocortex gyrencephaly, cytoarchitectonics, and chemoarchitectonics. (**A**) Perfusion fixed (4% PF) piglet brain illustrating the gyrencephalic cerebral cortex from a dorsal view. Anterior is at top. Gyri corresponding to the motor cortex, somatosensory cortex, and visual cortex are identified. (**B**) Nissl staining of a piglet brain sagittal section (40 µm) showing the prominent layer 5 of the motor cortex. (**C**) Nissl staining of a piglet brain sagittal section (40 µm) showing the prominent layer 4 of the somatosensory cortex. (**D**) Nissl staining of a piglet brain sagittal section (40 µm) showing the transition from the motor (left) to somatosensory (right) cortices. The motor cortex has a conspicuous layer 5 and inconspicuous layer 4 (agranular), while the somatosensory cortex has a prominent layer 4 and attenuated layer 5 (numbers identify cortical layers). (**E**,**F**) Piglet brain axial sections (anterior is at top) showing that the enzyme activity of cytochrome C oxidase (complex IV) is differentially enriched in regions of the cerebral cortex. (**G**) The inferior-most section with the centrum semiovale. Scale bars: (**B**,**C**), 105 µm; (**D**), 54 µm; (**E**) (same for (**F**,**G**)), 2 mm. See Appendix A for additional data on cytochrome C oxidase localization in the piglet neocortex.

**Figure 5 cells-12-02454-f005:**
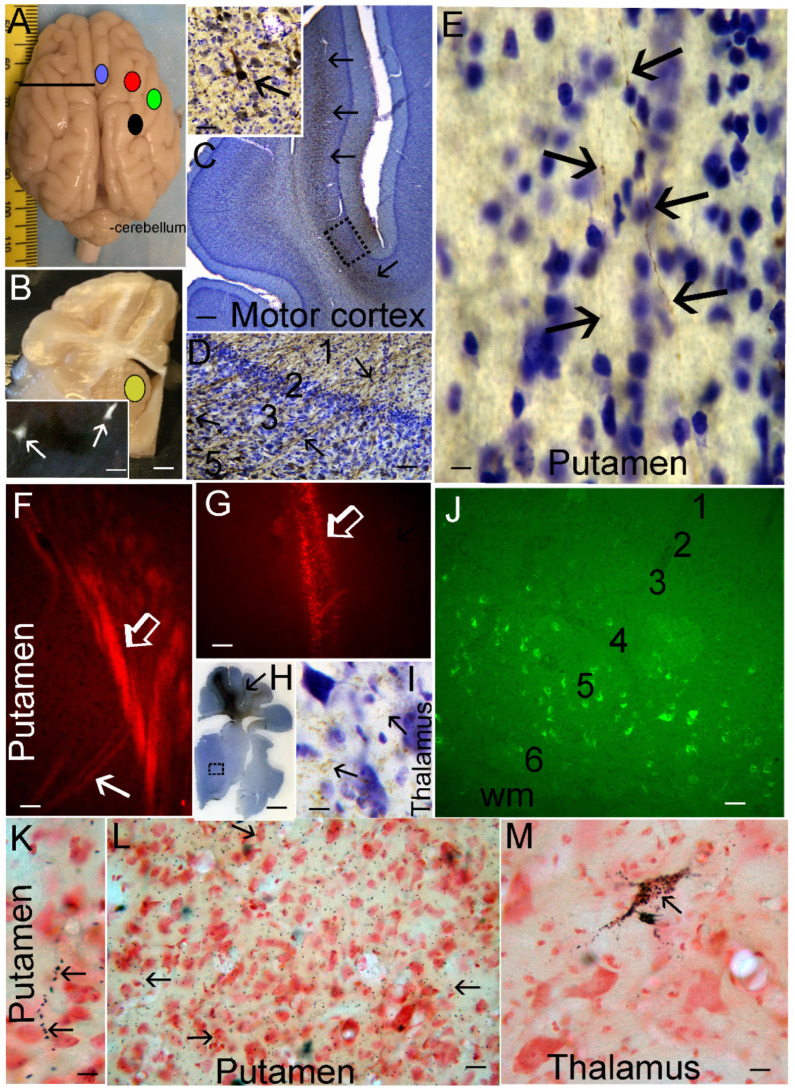
Neonatal piglet neocortical connectome. (**A**) Perfusion fixed (4% PF) piglet brain (gross) with color-coded buttons identifying locations of injection sites for AV-CFP (blue) in the right motor cortex, and AAV-eGFP (green), LV-RFP (red), and WGA-HRP (black) in the right somatosensory cortex. Black line in the left cerebrum identifies the approximate level of the brain slab shown in (**B**). (**B**) Piglet gross brain coronal slab identifying subcortical location of injection site for FluoroGold (FG, yellow) in the left striatum. Inset shows FG retrogradely labeled (white arrows) motor cortex corticostriatal projection neurons in layer 5 (direct fluorescence of FG). (**C**) CFP labeling of motor cortex (brown, antibody detection of CFP by immunohistochemistry with diaminobenzidine chromogen and blue Nissl counterstaining). Layer 5 neurons are positive (arrows) with inset showing a gigantocellular Betz cell (arrow) strongly positive for CFP. Area in hatched box is shown at higher magnification in (**D**). (**D**) Superficial layers of the lateral bank of the motor cortex gyrus showing numerous CFP-labeled apical dendrites of layer 5 neurons extending through layers 4, 3, 2, and 1 to the cortical surface (numbers identify cortical layers). (**E**) Single-axon resolution labeling (arrows) of CFP motor corticostriatal projections in the putamen. (**F**,**G**) RFP labeling of somatosensory corticostriatal projections to the putamen seen in the external capsule (open arrow) and white matter stria (solid arrow) in the lateral putamen (direct fluorescence of RFP). RFP labeling near the somatosensory cortex (**G**) injection site for LV-RFP (open arrow, direct fluorescence of RFP). (**H**) Somatosensory cortex injection site for LV-RFP (brown, arrow) seen by antibody detection of RFP. Box in ventral thalamus is shown in (**I**). (**I**) RFP anterograde labeling of fine somatosensory corticothalamic terminal field plexus (arrows) in the ventroposterolateral thalamic nucleus. (**J**) Corticocortical projections identified by eGFP-positive (direct fluorescence) retrogradely labeled neurons concentrated in layer 5 of the cingulate cortex after AAV-eGFP injection in the somatosensory cortex. (**K**) HRP-labeled single axon (arrows) in the lateral putamen (near external capsule) resulting from injection of WGA-HRP in the ipsilateral somatosensory cortex (neutral red counterstaining). (**L**) Within the central putamen, the HRP-labeled somatosensory corticostriatal terminal field is enriched (arrow, black dots represent individual presynaptic terminals). The putamen contralateral to injections of WGA-HRP in the somatosensory cortex was essentially negative, indicating negligible crossed corticostriatal projections. (**M**) WGA-HRP retrogradely labeled neuron (arrow) in the thalamic ventroposterolateral nucleus that projects to the somatosensory cortex. Scale bars (in µm, except for (**B**,**H**)): (**B**), 1.75 mm; (**B**) inset 30; (**C**), 293; (**C**) inset, 35; (**D**), 25; (**E**), 8.5; (**F**), 10; (**G**), 10; (**H**), 2.6 mm; (**I**), 7; (**J**), 44; (**K**), 12; (**L**), 21; (**M**), 10.5.

**Figure 6 cells-12-02454-f006:**
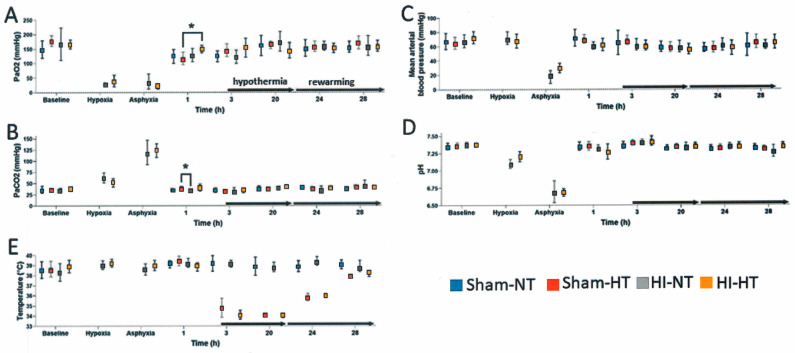
Physiology of HI and sham piglets with or without HT. Data are shown as means with 95% confidence intervals. Blood gas data are shown from 7 min of asphyxia. Temperature and blood pressure data are shown from 8 min of asphyxia. The first arrows along the *x*-axis show the beginning and maintenance of HT. The second arrows along the *x*-axis show rewarming. (**A**) Arterial partial pressure of oxygen (PaO_2_) varied across time (*p <* 0.001). (**B**) Arterial partial pressure of carbon dioxide (PaCO_2_) was interactively affected by time and treatment (*p *= 0.006). Mean arterial blood pressure (**C**), arterial blood pH (**D**), and core body temperature (**E**) were affected by treatment. For all parameters, group sizes were: sham normothermia-NT (*n* = 6), sham hypothermia-HT (*n* = 10), HI-NT (*n* = 8), and HI-HT (*n* = 10). * *p* < 0.05 was deemed significant in post-hoc tests. See Appendix A for additional hemodynamic data.

**Figure 7 cells-12-02454-f007:**
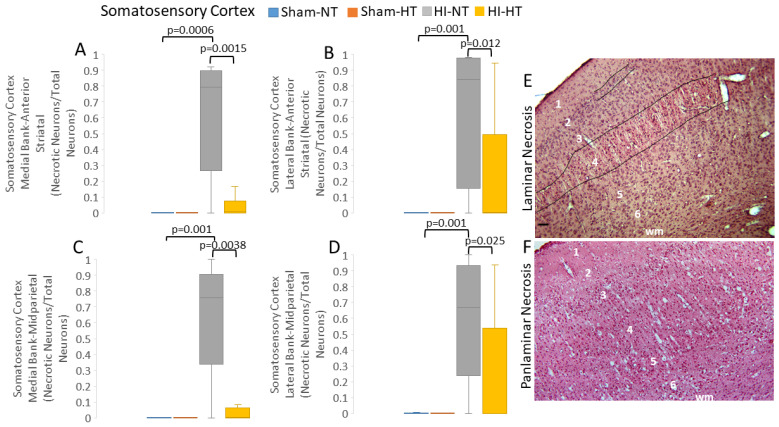
The primary somatosensory neocortex was protected in HI-HT piglets. Box plots show mean values with IQR and 5–95th percentile whiskers. Ischemic-necrotic neurodegeneration was infrequent or not detected in sham piglets. It was highest in HI-NT piglets and was significantly increased in the different regions of the somatosensory cortex compared to sham-NT piglets. Ischemic-necrotic neurodegeneration was significantly attenuated in HI-HT piglets compared to HI-NT piglets in the anterior parietal gyrus (at the anterior striatal level) medial bank (**A**), lateral bank (**B**), and in the mid-parietal gyrus (at a thalamic level) medial bank (**C**), and lateral bank (**D**). *p* < 0.05 was deemed significant in post-hoc *t*-tests. (**E**) H&E staining of the somatosensory cortex of an HI piglet showing clear selective laminar pathology in lower layer 3 and upper layer 4 and more subtle damage in layer 2 (numbers identify cortical layers), as delineated by the black lines. (**F**) H&E staining of the somatosensory cortex of an HI piglet showing overt panlaminar necrosis. See Appendix A for additional comparisons. Scale bar in (**E**) (same for (**F**)), 63 µm.

**Figure 8 cells-12-02454-f008:**
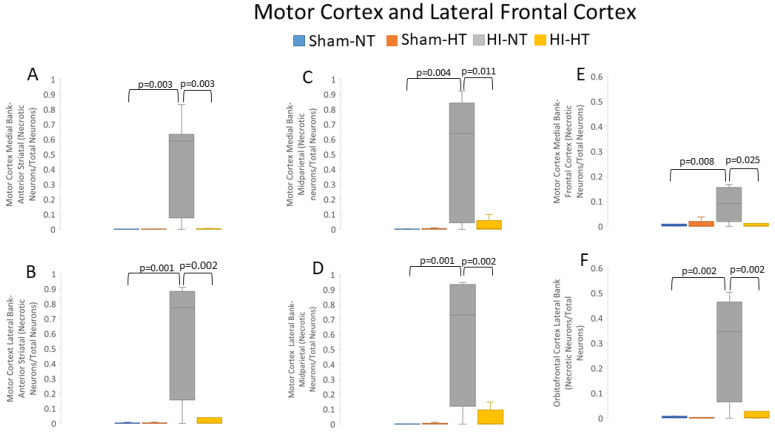
Motor cortex damage after HI varied regionally, protection in HI-HT piglets, and vulnerability of the orbitofrontal cortex. Box plots show mean values with IQR and 5–95th percentile whiskers. Ischemic-necrotic neurodegeneration was not detected, infrequent, or low in sham piglets. It was highest in HI-NT piglets. Ischemic-necrotic neurodegeneration was significantly attenuated in HI-HT piglets compared to HI-NT piglets in the anterior parietal gyrus (at the anterior striatal level) medial bank (**A**), lateral bank (**B**), and mid-parietal gyrus (at a thalamic level) medial bank (**C**), and lateral bank (**D**). The anterior-most region of the motor cortex in the anterior frontal cortex was less vulnerable than more posterior regions of the motor cortex (**E**). The lateral orbitofrontal region showed significant vulnerability in HI-NT piglets and protection with HT (**F**). *p* < 0.05 was deemed significant in post-hoc *t*-tests.

**Figure 9 cells-12-02454-f009:**
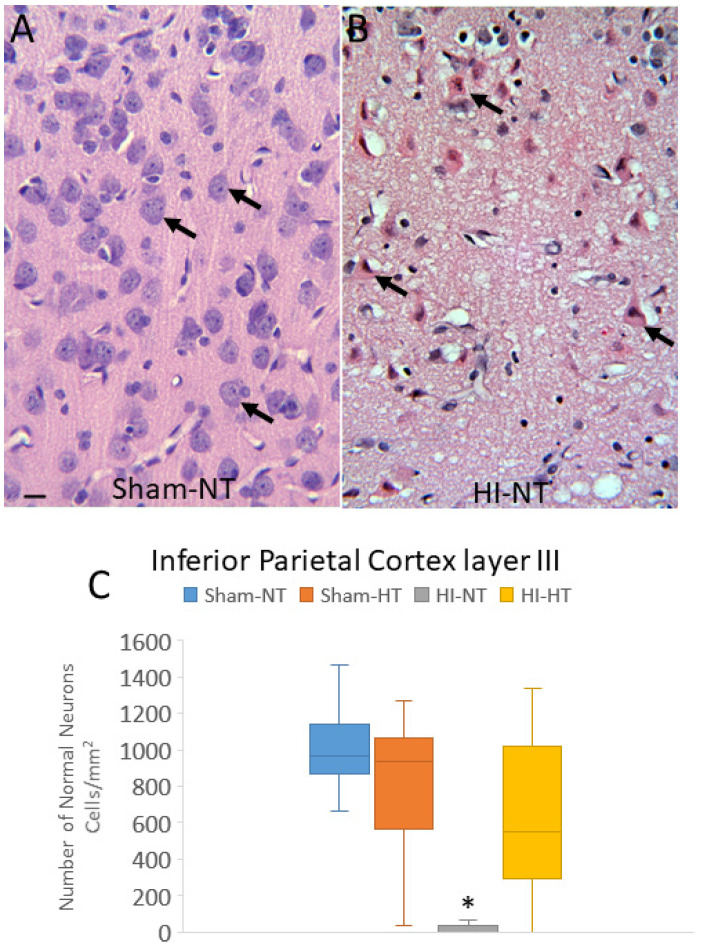
Layer III had prominent neuronal loss in the inferior parietal neocortex of HI-NT piglets. See Figure 2F for a low magnification image identifying the location of where the inferior parietal cortex was analyzed in each piglet. (**A**) In sham-NT piglets, H&E staining showed pristine undamaged neuronal cell bodies (arrows) and neuropil composition. Scale bar (same for (**B**)), 13 µm. (**B**) In HI-NT piglets, layer III was devastated with only residual ischemic-necrotic neurons (arrows) remaining in a field of spongiform neuropil. (**C**) HI-NT piglets had a severe loss of normal neurons in layer III of the inferior parietal cortex compared to sham-NT piglets (* *p* < 0.001). Normal neuron number was partly rescued in HI-HT piglets compared to HI-NT piglets (*p* < 0.001). Box plots show median values with IQR and 5–95th percentile whiskers. * *p* < 0.05 was deemed significant in post-hoc *t*-tests.

**Figure 10 cells-12-02454-f010:**
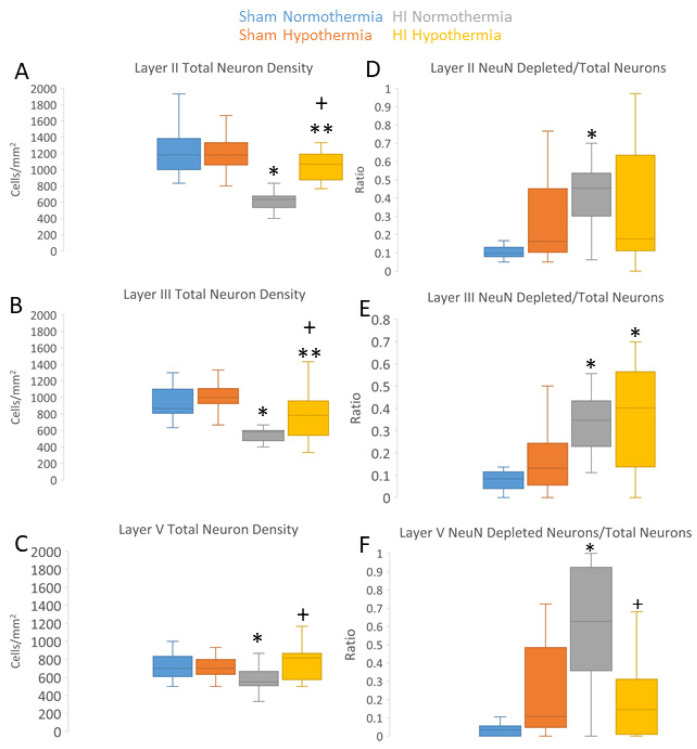
The somatosensory cortex in HI-NT piglets had differential laminar vulnerability and nuclear depletion of Rbfox3/NeuN immunoreactivity. Brain sections immunostained for Rbfox3 and CV-counterstained were used to count total neurons (**A**–**C**) and neurons depleted of nuclear Rbfox3 positivity (**D**–**F**) in layers II, III, and V of the somatosensory cortex. Box-and-whisker plots show the median and interquartile ranges. (**A**–**C**) In sham piglets, layer II had the most neurons. In HI-NT piglets, total neuron counts were reduced (* *p* < 0.001) compared to those in sham-NT piglets in layers II (**A**), III (**B**), and V (**C**). In HI-HT piglets, total neuron counts were reduced (** *p* < 0.01) compared to those in sham-HT piglets in layers II (**A**) and III (**B**). There was significant (+ *p* = 0.009) rescue of total neurons in HI-HT piglets compared to HI-NT piglets. (**D**–**F**) In HI-NT piglets, the ratios of Rbfox3-depleted-to-total neurons increased in all layers compared to those in sham-NT piglets (*p* < 0.001). In HI-HT piglets, the ratio of Rbfox3-depleted-to-total neurons increased in layer II compared to that in sham-NT piglets (*p* < 0.001). There was significant (+ *p* = 0.001) rescue of total neurons in HI-HT piglets compared to HI-NT piglets in layer V. *p* < 0.05 was deemed significant in post-hoc *t*-tests.

**Figure 11 cells-12-02454-f011:**
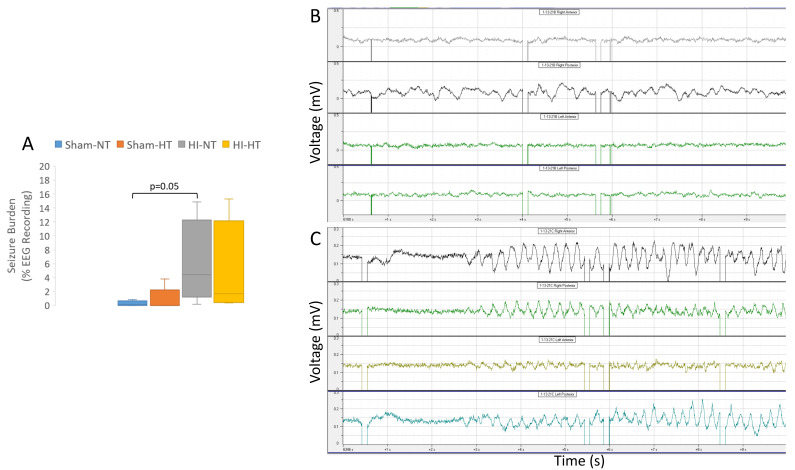
Encephalographic seizure assessment in neonatal piglets. (**A**) Seizure burden was greater in hypoxia-ischemia (HI)-normothermic (NT) piglets than in sham-NT piglets (*p* = 0.05). (**B**) Representative 10 s electrograms of neocortical activity in right anterior, right posterior, left anterior, and left posterior leads in a sham-NT piglet. EEG shows background activity of a variety of frequencies and amplitudes throughout all leads. No rhythmic waveforms are present. (**C**) cEEG of HI piglet with seizures showing the presence of epileptiform appearing as rhythmic spike-wave complexes (2–3 Hz) generalized in all four leads over 10 s.

**Figure 12 cells-12-02454-f012:**
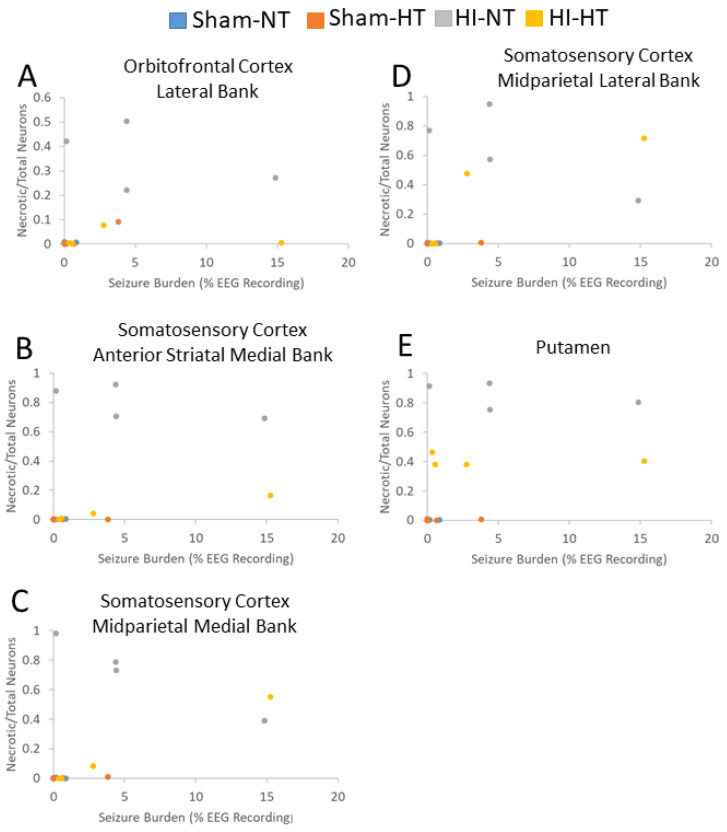
Piglets with greater seizure burden had more ischemic necrosis in the orbitofrontal cortex ((**A**), r = 0.48, *p *= 0.045), anterior (striatal level) somatosensory cortex ((**B**), r = 0.57, *p *= 0.015, *n* = 18), mid-parietal somatosensory cortex ((**C**), medial bank: r = 0.69, *p *= 0.002, *n* = 18; (**D**), lateral bank: r = 0.62, *p *= 0.006), and putamen ((**E**), r = 0.60, *p *= 0.009). Normothermia (NT), hypothermia (NT), hypoxia-ischemia (HI).

## Data Availability

Contact L.J.M.

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
