# Peer review of "Hypothermic Protection in Neocortex Is Topographic and Laminar, Seizure Unmitigating, and Partially Rescues Neurons Depleted of RNA Splicing Protein Rbfox3/NeuN in Neonatal Hypoxic-Ischemic Male Piglets"

_cells, 2023, doi:10.3390/cells12202454_

Round 1

Reviewer 1 Report (Previous Reviewer 1)

I thank the authors for their edits. The manuscript has been improved but I still have concerns about some of the figures, in particular:

- Figure 6 is still essentially illegible, especially panel G. Could lines be added to connect the points to make the panels easier to follow? I also strongly recommend making the panels much larger and/or moving the less essential panels to supplemental, again to allow them to be larger.

- The y-axis legends in figure 7 panels A-D are unreadable due to a combination of small text size and low image quality.

- Why do all the box plots have such a large gap between the y-axis and the Sham-NT data? This also hinders interpretability.

- The colours and group names in Figure 12 could be matched to those in the other figures for better continuity.

Some minor edits to the text are needed - e.g. "deceased" line 225 and "NT-NT" line 769.

Author Response

Thank you for taking the time to skillfully review the manuscript again.

Reviewer 1

-Figure 6 is still essentially illegible.

Response: We have remade figure 6 to correct the resolution problem. Key graphs were selected from the larger figure and shown in the new Figure 6 with better legibility. A larger figure with all the graphs is now shown with better resolution in the supplementary material as Figure 1.

                -The y-axis legends in figure 7 are unreadable.

Response: We have remade figure 7 with larger font for the y-axes of all the graphs.

                -box plot gaps

Response: The box plot design was kept the same (with gaps) for better legibility after figure size reduction.

                -The colours and group names in Figure 12.

Response: We have entirely remade figure 12 for group color and name consistency with other figures.

                -Minor edits

Response: The typos were corrected.

Reviewer 2 Report (New Reviewer)

The manuscript by Primiani et al. is a revised one previously unseen by this Reviewer. Hence, issues may be raised that were already addressed by other Reviewers in previous round(s) of submission, apologies for that.

This is an industrious preclinical study concerned with the pathology and pathophysiology of neonatal hypoxic-ischemic encephalopathy that aims on one hand the detailed quantitative analysis of the neuronal damage in specific neocortical areas induced by hypoxic-ischemic stress, and on the other hand the also quantitative description of neuroprotection elicited by the gold-standard neuroprotective treatment of HIE patients: therapeutic hypothermia (HT).

The authors, building on extensive previous experience of the research group with the model, establish a long-survival (2-7 days) translational newborn pig HIE model with continuous EEG monitoring to establish seizure burden. Also, in a set of preliminary functional anatomy studies, they painstakingly identify and characterize the later assessed neocortical regions in the gyrencephalic piglet brain. The main source of data is the neuropathological analysis in H&E stained brain sections, and in RBFOX3/NeuN (NeuN) immunohistochemistry slides counterstained with cresyl-violet (Nissl-stain).

The work is very interesting and furthers our knowledge regarding the pathophysiology of neuronal damage / neuroprotection by HT in this important translational gyrencephalic HIE model, however, a number of issues need to be addressed before the full merit of the work can be appreciated.

Major concerns:

1.       The abstract states (line 28): “HI-NT piglets had reduced normal/total neuron ratio and increased 28 ischemic-necrotic/total neuron ratio relative to sham-NT and sham-HT piglets with differing severities in anterior and posterior motor, somatosensory, and frontal cortices.” However, I failed to find any statistical comparison showing difference among the different cortical regions comparing the values obtained from the HI-NT group.  Such major difference is not apparent from the figures, especially that variability appears to be very high in this group as expected. Please explain, and to maintain the statement supporting statistical data should be reported in the manuscript.

2.       The abstract states (line 32): “Rbfox3 immunoreactivity identified cortical neurons as:  Rbfox3-positive/normal, Rbfox3-positive/ischemic-necrotic, and Rbfox3-depleted.” This is shown in Figure 3 and Results 2.6. The “Rbfox3-depleted” neurons are described as having normal morphology on the Nissl stain. This is very intriguing: does this mean that there are no NeuN depleted dying/dead neurons at all? This would imply that this response would “save” neurons – NeuN immunopositive cells may die or may survive but all NeuN depleted cells survive? Please explain, and state in the manuscript explicitly if ischemic/necrotic NeuN-depleted neurons were indeed absent.

3.       The Methods state (line 263): “Sham procedure piglets were euthanized as time matches 263 for HI piglets throughout the 2 to 7-day period.” This statement appears to be in contradiction with Supplementary Figure 1 showing survival in Sham NT minimum 4 days while in HI-NT more than half of the group survived only 2 days. Please explain and dissolve this contradiction.

4.       Statistics (lines 479-480). What is the rationale of the choice of t-tests? What is the real difference between this and the use of the Fisher’s LSD post hoc test?

5.       Following on the #1 concern, could the authors demonstrate a regional difference in HT neuroprotective potential?

6.       The Discussion could be extended to discuss other potential causes of reductions in NeuN immunoreactivity reported previously such as axonal injury, change in immunogenicity etc. Also the prominent cytoplasmic presence of NeuN could be discussed.

7.       The careful, major revision of the title and the abstract is recommended to better advertise the major messages of the manuscript.

Minor (also stylistic) points:

line 22: continuous electroencephalography (cEEG) instead of “encephalography”

line 23: “overnight” could/should be omitted – duration of HT should be given precisely or not disclosed at all in abstract

line 125: “…serially cut on a sliding microtome into 40 μm thick floating sections in sagittal, horizontal, and coronal planes”  perhaps “either” could be included in front of the planes?

line 166: concentration of injected tracers is given. Could you give also the volume of injection for reproducibility?

line 266: “All piglets achieved uniform fixation.” perhaps a passive sentence would be better.

line 296: (Figure 3) – On the figure cortical layers are labeled with Arabic numerals (1-6), however, in the legend and also virtually everywhere in the text, the more customary Roman numerals (I-VI) are used.  This is also true for Figure 4. For clarity, please unite labeling

line 300: Western instead of western is recommended

line:332: Figure 4 does not seem to support the message of the paper too much, Nissl-stained slides were not used for neuropathology, the neuropathology results are not correlated with cyctochrom oxidase immunohistochemistry. This Figure could be safely omitted, especially in light of the plethora of Figures and the complexity of the manuscript.

line 539: Figure 6 – the visibility of the figure is quite poor, not legible below 200% magnification. Moreover, there are unexplained black lines in Panels F&G. Please explain and revise. Compared to its present form, a table would be preferable.

line 619-622: There is a confusion created with group abbreviations, non-existent  HT-HT and HT-NT groups are being named here. Please clarify the text.

line 769: non-existent NT-NT group creates confusion. Please clarify the text.

line 770: “Normal piglet EEG patterns showed activity of various frequencies, primarily high (>10 Hz), and low voltage (0.1-0.2 mV)” 0.1 mV = 100 uV waves are most definitely NOT low voltage on EEG. Please correct the sentence.

 line 981: Previous instead of “Pervious”

Supplementary Figures 3, 5, 6, 7: the vertical (y) axis range should be made the same in all panels in order to better show the differences among the different regions. Also, the blue hue used for the boxes of the Sham NT group is slightly different (darker) than shown in the legend (and the other figures of the manuscript)

Author Response

Thank you for reviewing our manuscript so carefully and attentively. You really made great suggestions..

Reviewer 2

  1. The abstract and statistical comparisons.

Response: We have now made the statistical comparisons of the groups more thorough and apparent, with particular focus on the HI-NT group. The revised Figure 7 shows the comparisons of sham-NT piglets with HI-NT piglets. We have made two new tables shown in the Supplementary Material (Table 1 and table 2) that identify p values.

  1. The abstract and Rbfox3 immunoreactivity.

Response: We have now clarified the data on the Rbfox3 immunoreactivity in the Results and in the Discussion. We also explicitly identify and show in Figure 3C (inset) that degenerating neurons could be Rbfox3 positive.

  1. Methods and sham procedure piglet euthanasia.

Response: We have clarified when the sham piglets were euthanized as paired with HI piglets.

  1. Statistical choices.

Response: In the Methods (Statistical analysis), we describe our rationale for using t-tests.

  1. Could the authors demonstrate regional differences on HT neuroprotection.

Response: Yes, HT neuroprotection potential was regionally differential. We point this out in the descriptions of Figure 7 and Figure 8 results.

  1. The discussion could be extended to discuss the NeuN results.

Response: Because of the existing length of the Discussion, we did only a brief extension of the discussion of the NeuN results. We included a new reference (154) on NeuN changes in cerebral ischemia.

  1. Revision of the title and the abstract.

Response: We made major revisions to the Abstract, particularly regarding the conclusions. We did a change to the title. We also changed the title in the last version of the manuscript in accordance with a reviewer comment.

Minor points

  • We corrected to electroencephalography.
  • We omitted overnight.
  • We edited the section cutting sentence.
  • We stated the injection volumes.
  • We used passive tense for sentence.
  • We made many changes to balance the consistent use of Arabic versus Roman numerals for the cortical layer identification. Changes to figures and text were done.
  • We now use Western instead of western.
  • Figure 4. We kept Figure 4 in the manuscript because we deem it vital to the paper. It shows the gyrencephaly of the piglet neocortex in full measure and our ability to parcel it into regions. The parcellation is critical because the neuropathology described in the paper is regionally specified. Moreover, we used the Nissl staining as a counterstain for the NeuN.
  • Figure 6 visibility. We have remade figure 6.
  • We have corrected the animal group abbreviation typos.
  • Line 770. We edited this sentence regarding the EEG voltage.
  • Typo was corrected.

Reviewer 3 Report (New Reviewer)

Primiani and colleagues provided a very detail characterisation study on a novel model of hypoxic-ischemic in newborns using male piglets. In their long manuscript, they showed detailed examination of the neuronal loss in various brain regions based on Nissl, H&E, and Rbfox3 (NeuN) staining. Using various viral and chemical tracers, they provided some anterograde and retrograde tracing data. They also provide biochemical and electrophysiological data.

Major concerns:

1-(Fig 2 and manuscript esp. lines 389-411) Authors provided 4 characteristics of neuronal injury based on H&E staining: 1-normal neurons, 2-ischemic-necrotic degenerating neurons, 3-apoptosis-necrosis continuum degenerating neuron, and 4-apoptotic neurons. However, there are a couple issue with this classification. Firstly, unsure about the resolution that can be obtained solely from H&E staining for 4 classes, especially the apoptosis-necrosis continuum degenerating neurons class. Secondly, how important are these distinctions in regards to the pathophysiology? Thirdly, since the manuscript is rather long, it seems unnecessary  and wordy to add the extra ‘ischemic’ and ‘degenerating’ words? Fourthly, (line 1018-1034) Based on these characteristics of using only H&E staining, the discussion section seems over interpreted.

2-Several of the images provided are of different orientation making it difficult to compare. For example, Fig 3A (sham) and is used to compare Fig 3C (HI-NT), but Fig 3A showed cortical layers I-VI and white matter whilst Fig 3C only showed cortical layers I-IV and another part of the motor cortex . Therefore in fig 3C and D, authors need to show similar orientation and magnification as fig 3A and B so readers can compare it themselves. Note space is not an issue with this journal.

3-Avoid speculation. In line 309-310, suggested that there was ‘partial rescue of RBfox3 positivity in other neurons ..’ as it could also be not injured as not all HI causes the same injury, even when the exact experimental protocol are used.

4-Figure concerns. Fig 4. Need to provide higher magnification of staining for panels E –G. Fig 7E and F seems different in H&E staining quality. Also, require control tissue and higher magnifications of region to show the presence and absence of necrosis. Fig 9. Need to have lower magnifications to indicate the depicted regions were similar regions that were analysed. Fig 12. If plot the correlation lines for panel A and B, it suggests there is decrease in necrosis with increase seizure burden for HI-NT (red), but increase in necrosis with increase seizure burden for HI-HT (blue). Is this what was expected? Also, should the r value not be negative if it has a negative slope?

5-Statistical concerns. (Line 460 and 769) P value =0.05 is not significant so cannot say it’s significant. May need to use 3 dpi for clarity. Need to briefly explain why some data were analysed with ANOVA or Kruskal Wallis or Mann Whitney? Why use Spearman and not Pearson correlation? (line 480) What ANOVA and t-tests used?

6-Terminology concerns: (line 496, and others) ‘Attenuated’ is an inappropriate term to use as it suggest something caused this region to be reduced when it is normally this low level. (line 506) Maybe ‘Genetic’ should be changed to ‘viral’ or similar. (line 509) Change ‘infected’ to ‘transfected’ to be more appropriate. (line 511) Maybe change ‘human-like Betz’ to ‘human Betz-like’ neurons. (line 706) Avoid using ‘pristine’ word.

7-(line 529-533) Important to know which group of animals contributed to the 85% completion rate.

8-Fig 11. There are the presence, albeit low, of some seizure burden in sham groups. Are these animals experiencing seizure?

9-(First paragraph of discussion) (lines 819-820): incorrect statement as negative slope especially for fig 12A and B, and that fig 12C and D are plateau, suggesting not more seizures causes more neuronal loss. (Line 822): How about the data from fig 10?  (Line 828-830): Could the loss of neurons rather than specific depletion of Rbfox3/NeuN be the cause for what was observed?

10- (Discussion) Should include some discussion on other piglet hypoxic-ischemic models by others such as Thoresen (PMID: 7603788), Robertson (PMID: 33262073) and Henriksen (PMID: 26068784) groups.

 3-The supplementary videos were not available.

Minor concerns:

1-(Title) Suggest put NeuN next to RBFOX3 in the title as more familiar to most readers.

2-(lines 143, 167,186, 193) A drilled hole should not be considered as a craniotomy as no section of skull was removed.

3-(lines 155-169) What was the volume of viral vector and tracers used?

4-(line 235) Was heating blanket used to maintain normal body temperature?

5-Fig.2. should provide bregma or equivalent coordinates for others to reproduce.

6-Fig 6. Should use the same colors for the different groups as the rest of the graphs in the manuscript for consistency. Fig 6G. Why are the stars and brackets at the bottom of the graph?

7-Spelling errors. For example, line 619: HT-HT group, line 769 NT-NT group.

8- (line 690-703) Was seizing piglets that died before 7 days included in any of the analysis?

9-Fig. 10. Should use brackets to indicate which groups are being compared, similarly to other figs in the manuscript (e.g. Fig 8).

Author Response

Thank you for reviewing the manuscript so carefully and astutely. 

Reviewer 3

  1. Degenerating neuron classification

Response: We have developed and used this classification of neurodegeneration for a long time. It was developed originally using Nissl and TUNEL staining and also electron microscopy. We believe it to be validated. We have extended this classification to neurodegeneration in piglet using H&E staining. We provide references to support this classification (54, 56, 66, 69).  We have also used this classification of neurodegeneration on neonatal rodent HI models (Northington et al., 2007 Neuroscience).

  1. Image orientation

Response: We have made a new figure for Figure 3 (Supplementary Material Figure 3) The images in this figure are matched and labeled very well to facilitate ease of comparison.

  1. Avoid speculation.

Response: We clarify this statement with data showing that the injured neurons retain NeuN immunoreactivity in a mis-localized and aggregated form that is distinct from the NeuN depleted neurons.

  1. Figure concerns, Fig. 4, Fig. 7, Fig. 9, and Fig. 12.

Response: For Figure 4, we have made a new figure shown as Supplementary Material Figure 4 that illustrates the cytochrome C oxidase staining at high magnification. With Fig. 7, panel F and E appear different because with severe damage, there is intense eosinophilia and that is why 7E appears so pink. For Figure 9, we now refer in the text to Figure 2 where the levels of analysis are shown at low magnification. In Figure 12, the y-axis is ratio of ischemic-necrotic/total neurons and the x-axis is seizure burden. In the HI animals, there is positive correlation because both values increase in the same direction, so the r is positive.

  1. Statistical concerns.

Response: We made the correction regarding P value = 0.05. For Figure 12, we used the Pearson correlation because we were examining relationships between only two variables.  We identify the type of ANOVA used and the type of t-test used.

  1. Terminology concerns.

Response: The suggested changes were made.

  1. Which group of animals contributed to the 85% completion rate.

Response: Animal completion and attrition are described in the Supplementary Material.

  1. There is presence of some seizure burden in the shams.

Response: The reviewer is correct, and we describe this in the results.

  1. First paragraph of Discussion.

Response: We have clarified the Figure 12 graphs that show that the higher the seizure burden the higher the value of ischemic-necrotic neurons. We also clarified the NeuN results by showing that ischemic-necrotic or continuum degenerating neurons have retained NeuN immunoreactivity that is abnormally mis-localized and aggregated. The nuclear depletion appears to be a distinct phenomenon. 

  1. Discussion of other piglet models.

Response: In the discussion, we do cite work on other large models of neonatal HI because it was a comment in a prior review. We do cite Dr. Thoresen’s work extensively in the introduction.

3- supplementary videos.

Response: The link for the videos is provided in the revised manuscript and in the supplementary material file.

Minor concerns:

  1. We added NeuN to the title.
  2. We replaced craniotomy with burr holes.
  3. We added the volumes of the tracers used.
  4. Yes, a heating blanket was used to maintain normal body temperature.
  5. Bregma is now indicated in Figure 4.
  6. Figure 6 has been remade with colors for the different groups as the rest of the graphs in other figures.
  7. Typos have been corrected.
  8. Yes, all piglets, regardless of early euthanasia due to seizures, were included in the analysis.
  9. Because the use of brackets made the figure very busy and less clear, we used symbols.

Round 2

Reviewer 1 Report (Previous Reviewer 1)

This version is fine.

Reviewer 2 Report (New Reviewer)

The authors have addressed the concerns and modified the manuscript accordingly. Congratulations!

Reviewer 3 Report (New Reviewer)

All my concerns have been addressed, so there are no further concerns regarding this manuscript for publication.

This manuscript is a resubmission of an earlier submission. The following is a list of the peer review reports and author responses from that submission.

Round 1

Reviewer 1 Report

The authors of the manuscript entitled “Hypothermic Protection in Neocortex is Topographic and Laminar, Seizure Unmitigating, and Partially Rescues Neurons 3 Depleted of RNA Splicing Protein RBFOX3 in Neonatal 4 Hypoxic-Ischemic Piglets” provide an exceptionally detailed neuropathological assessment of regional- and layer-specific neoncortical hypothermic neuroprotection in a piglet model of neonatal hypoxia-ischemia (HI). They find that neonatal piglet neocortex has a spatial vulnerability to HI and seizures and spatial response to hypothermia (HT), with a short period of HT not reducing seizure burden, which was associated with a novel  Rbfox3 immunophenotype. As expected from this group the animal model work and in particular the neuropathological assessments, are meticulous. However, significant methodological limitations should receive more attention, and the presentation of the data could be improved.

Major comments

·      Both the abstract and introduction are long and dense, with much of that information probably more suitable to the discussion. I would  perhaps expect the abstract to be half the length it currently is to more succinctly introduce the reader to the work. The introduction also includes a lot of background that the reader might not need, but Rbfox3/NeuN – a critical aspect of one component of the study – is only very briefly introduced right at the end.

·      The data presentation is difficult to navigate as a reader. Why do the results switch from mean+/-95% CI to box plots? Even if this is due to assumed data distribution, with n=6-10 one can never robustly determine the underlying distribution. Plotting neuropathology counts as mean with 95% CI dramatically limits the information that is being conveyed, especially when a those limits cross 0 (which therefore includes a potential result that is impossible). As the group sizes are so small, all the plots should just show all the data. The graphs in figure 6-8 are also essentially impossible to read or interpret because they are so small.

·      Discussion around the length of HT and comparison to another non-primate gyrencephalic model of HIE in the fetal sheep is  conspicuous in its absence. In fetal sheep, HT may suppress early (but not late) seizure burden, and there is a dose response with HT where 72h appears to be close to optimal for a single acute asphyxia event, and is more protective than shorter periods (e.g., https://www.ncbi.nlm.nih.gov/pmc/articles/PMC6585539/). Could the absence of certain potential neuroprotective effects of HT be due to a suboptimal HT exposure?

·      Regarding the discussion of regional susceptibility to HI and HT in humans, more nuance regarding the type of event is also needed. For instance, patterns of injury in acute sentinel events such as modelled here versus more chronic or intermittent injuries that are thought to result in watershed injury patterns and are less likely to respond to HT.

·      HIE is a clinical diagnosis, so when discussing preclinical models (such as in rodents), this should be referred to HI rather than HIE.

·      As several rodent studies suggest that biological sex alters the cell death spectrum as well as response to hypothermia (https://pubmed.ncbi.nlm.nih.gov/25293493/, https://pubmed.ncbi.nlm.nih.gov/32616806/), the use of only male animals could potentially limit the translatability of these results. The use of a single sex is also not justified. I would strongly recommend that “male piglets” be added to the title, and a clearer discussion regarding whether females might be expected to have similar (or different) results be added. In general, the limitations should not just be listed, but should be used to provide more context to the results.

·      For the neuropathology data, were t-tests essentially performed as post-hoc tests after ANOVA? If so, why not a test that better accounts for multiple comparisons, as done for the physiologic data? The target p-value level and statistical analysis software are also not mentioned. It would also be helpful to include the statistical approach in each of the figure legends.

N/A

Reviewer 2 Report

Dear authors,

This paper is aiming to describe a swine model of hypothermic protection upon hypoxic-Ischemia. This article presented data of various aspects. However, data are descriptive and toward the end, the article did not explain and concentrate on presenting results from all the experiments performed. Regrettably, the decision for this manuscript would be a rejection.

There are some points for your consideration:

1. There are 5 questions raised in the introduction but none of those are being answered properly in the discussion or a brief summary at the end of results after all the experiments.

2. Descriptive data need to be trimmed to only present important points of the article.

3. All image data should be on the same scale presumably, unless you have the explanation to present in different scale. Please also annotate all the information labeled on images for completeness.

4. Most of the figures are too small for publication and for reading. Please just present the important ones and set the rest in the supplements.

5. All performed experiments should have clear statement of results. Please try to link to answering your questions in the introduction.

6. Please show results of HI-HT for all your experiments. This is your key of the whole article, not just show HI-NT to compare with shame-NT.